# Intravenous and intracranial GD2-CAR T cells for H3K27M+ diffuse midline gliomas

Michelle Monje[1,2,3,4,5,6 ✉], Jasia Mahdi[1,3], Robbie Majzner[2,3], Kristen W. Yeom[1,4,7], Liora M. Schultz[2,3], Rebecca M. Richards[2,3], Valentin Barsan[2,3], Kun-Wei Song[1,3], Jen Kamens[2,3], Christina Baggott[3], Michael Kunicki[3], Skyler P. Rietberg[3], Alexandria Sung Lim[3], Agnes Reschke[2,3], Sharon Mavroukakis[3], Emily Egeler[3], Jennifer Moon[3], Shabnum Patel[3], Harshini Chinnasamy[3], Courtney Erickson[3], Ashley Jacobs[3], Allison K. Duh[4], Ramya Tunuguntla[3], Dorota Danuta Klysz[3], Carley Fowler[3], Sean Green[2], Barbara Beebe[3], Casey Carr[3], Michelle Fujimoto[3], Annie Kathleen Brown[3], Ann-Louise G. Petersen[3], Catherine McIntyre[8], Aman Siddiqui[8], Nadia Lepori-Bui[8], Katlin Villar[8], Kymhuynh Pham[8], Rachel Bove[8], Eric Musa[8], Warren D. Reynolds[3], Adam Kuo[3], Snehit Prabhu[3], Lindsey Rasmussen[9], Timothy T. Cornell[9], Sonia Partap[1], Paul G. Fisher[1], Cynthia J. Campen[1], Gerald Grant[4], Laura Prolo[4], Xiaobu Ye[10], Bita Sahaf[3], Kara L. Davis[2,3], Steven A. Feldman[3], Sneha Ramakrishna[2,3 ✉] & Crystal Mackall[2,3,11,12 ✉]

H3K27M-mutant diffuse midline gliomas (DMGs) express high levels of the disialoganglioside GD2 (ref. 1). Chimeric antigen receptor-modified T cells targeting GD2 (GD2-CART) eradicated DMGs in preclinical models[1]. Arm A of Phase I trial no. NCT04196413 (ref. 2) administered one intravenous (IV) dose of autologous GD2-CART to patients with H3K27M-mutant pontine (DIPG) or spinal DMG (sDMG) at two dose levels (DL1, $1 \times 10^6$ kg$^{-1}$; DL2, $3 \times 10^6$ kg$^{-1}$) following lymphodepleting chemotherapy. Patients with clinical or imaging benefit were eligible for subsequent intracerebroventricular (ICV) intracranial infusions ($10–30 \times 10^6$ GD2-CART). Primary objectives were manufacturing feasibility, tolerability and the identification of maximally tolerated IV dose. Secondary objectives included preliminary assessments of benefit. Thirteen patients enroled, with 11 receiving IV GD2-CART on study ($n = 3$ DL1 (3 DIPG); $n = 8$ DL2 (6 DIPG, 2 sDMG)). GD2-CART manufacture was successful for all patients. No dose-limiting toxicities occurred on DL1, but three patients experienced dose-limiting cytokine release syndrome on DL2, establishing DL1 as the maximally tolerated IV dose. Nine patients received ICV infusions, with no dose-limiting toxicities. All patients exhibited tumour inflammation-associated neurotoxicity, safely managed with intensive monitoring and care. Four patients demonstrated major volumetric tumour reductions (52, 54, 91 and 100%), with a further three patients exhibiting smaller reductions. One patient exhibited a complete response ongoing for over 30 months since enrolment. Nine patients demonstrated neurological benefit, as measured by a protocol-directed clinical improvement score. Sequential IV, followed by ICV GD2-CART, induced tumour regressions and neurological improvements in patients with DIPG and those with sDMG.

Chimeric antigen receptors (CARs) couple an antigen-binding domain to T cell signalling domains to redirect T lymphocytes to cancer cells expressing a target of interest. Autologous CAR T cells have mediated impressive results in refractory B and plasma cell malignancies[3–7], but have not demonstrated high rates of sustained antitumour effects in solid cancers or brain tumours[8–15], with the exception of promising responses in a recent trial of GD2-CAR T cell therapy for neuroblastoma[16]. Differential rates of activity between liquid cancers and solid/brain tumours may relate to a dearth of safe targets with high homogenous expression, inadequate T cell trafficking and/or T cell dysfunction induced by the tumour microenvironment.

H3K27M-mutated diffuse midline gliomas (DMGs) chiefly occur in children and young adults and originate in midline structures of the nervous system. Patients with pontine DMG (also called diffuse intrinsic pontine glioma, DIPG) have a median overall survival of 11 months from diagnosis, and 5 year overall survival below 1% (refs. 17–19). Patients with DMGs outside of the brainstem, including the spinal cord, have a median overall survival of 13 months[18]. DIPG is the most common cause

of death due to brain cancer in children, with palliative radiotherapy the standard of care. Cytotoxic chemotherapy has not improved outcomes to date[20]. Although targeted therapies and immuno-oncology strategies have begun to show early promise[21–23], outcomes remain dismal.

We discovered high, uniform expression of GD2, a disialoganglioside, on H3K27M+ DMG cells and demonstrated that intravenous (IV) administration of chimeric antigen receptor-modified T cells targeting GD2 (GD2-CART) eradicated established DMGs in patient-derived orthotopic xenograft mouse models[1]. We and others also demonstrated increased potency and decreased systemic inflammation following intracerebroventricular (ICV) administration of CAR T cells compared with IV administration in preclinical brain tumour models[24–27]. These data provided rationale for this first-in-human/first-in-child phase 1 clinical trial (NCT04196413, a phase 1 clinical trial of autologous GD2-CART for DIPG and spinal diffuse midline glioma (sDMG)). We previously reported antitumour activity and correlative findings from the first three patients treated on NCT04196413 arm A at dose level 1 (DL1), and a fourth who enroled on trial but received therapy through a single-patient-compassionate investigational new drug (IND) application[2]. Here we report final clinical results following enrolment completion of arm A, which demonstrate the tolerability of $1 \times 10^6$ GD2-CART cells per kg administered intravenously (DL1) followed by sequential ICV infusions, tumour regressions and, in some cases, sustained antitumour effects in patients with DIPG and sDMG.

## Trial design

Eligible patients were 2–30 years of age, with biopsy-confirmed H3K27M-mutated DIPG or sDMG, who had completed standard frontline radiotherapy at least 4 weeks before enrolment, were not receiving corticosteroid therapy and had acceptable performance status (Lansky or Karnofsky score 60 or above). Although patients typically exhibited bulky disease in the brainstem or spinal cord, those with bulky disease involving the thalamus or cerebellum were not eligible due to increased risk of toxicity of GD2-CART observed with these tumour locations in murine models[1]. Patients with clinically significant dysphagia (as an indicator of significant medullary dysfunction) were also ineligible. Detailed eligibility and exclusion criteria are given in Supplementary Methods 1. The clinical trial protocol was approved by the Stanford Institutional Review Board and registered with ClinicalTrials.gov (NCT04196413). Informed patient or parent consent and child assent were obtained.

Primary objectives were the determination of the feasibility of manufacturing, assessment of safety and tolerability and identification of a maximally tolerated and/or recommended Phase 2 dose of IV GD2-CART following lymphodepleting chemotherapy in this population. Secondary objectives (detailed in Supplementary Methods 2, section 1) included preliminary assessment of benefit as measured by radiographic response on magnetic resonance imaging (MRI) and by clinical improvement in neurological dysfunction. IV doses were escalated using a 3 + 3 design, based on the occurrence of dose-limiting toxicities (DLTs) possibly, probably or most likely attributed to IV GD2-CAR T and occurring within 28 days following infusion. DLTs comprised any grade 5 toxicity, grade 4 cytokine release syndrome (CRS), grade 4 neurotoxicity lasting at least 96 h, new grade 3 neurotoxicity lasting at least 28 days, grade 4 neutropenia or thrombocytopenia lasting more than 28 days and grade 3 or greater non-haematologic toxicity, with the exceptions of those detailed in Supplementary Methods 2, section 5.4.5. Because sDMGs are rare, the protocol allowed safety in the DIPG cohort to inform dose escalation for patients with sDMG but, given the risk for neurotoxicity related to the location of DIPG, safety in sDMG patients did not inform dose escalation for patients with DIPG.

In December 2020, a protocol amendment added a new secondary objective to evaluate safety and assess clinical benefit to patients treated with IV GD2-CAR T cells followed by ICV administration of GD2-CAR T cells (Supplementary Methods 2). Patients were eligible to receive second or subsequent IV or ICV infusions if they showed complete response, partial response, minor response or stable disease radiographically on MRI, or had evidence for clinical benefit from preinfusion baseline according to the protocol-specified clinical neurological examination, and if at least 28 days had elapsed since the initial GD2-CART infusion or 21 days from subsequent infusions, circulating CAR T cell levels were below 5% and toxicity had resolved to below grade 2. Lymphodepletion was not administered before ICV infusions. All subsequent infusions were delivered intracerebroventricularly.

Following enrolment on the trial, no additional, non-protocol-directed chemotherapies, molecularly targeted therapies or immunotherapies were allowed. Local therapies such as intratumoural cyst drainage or re-irradiation were allowed; if re-irradiation was administered, subsequent infusions resumed after 4 weeks from the end of re-irradiation.

## GD2-CART manufacture

GD2-CART were manufactured in the automated CliniMACS Prodigy with IL-7 (Miltenyi Biotec, 12.5 ng ml⁻¹) and IL-15 (Miltenyi Biotec, 12.5 ng ml⁻¹) in the presence of dasatinib (Extended Data Fig. 1a and Supplementary Methods 3), and cryopreserved at day 7 of culture. The GD2-CAR was encoded by a retroviral vector encoding an iCasp9 domain (Bellicum Pharmaceuticals, Inc.), and GD2.4-1BB.CD3z CAR separated by a P2A ribosomal skip sequence (Extended Data Fig. 1b).

## Toxicity monitoring and management

Patients received standard supportive care for seizure prophylaxis, immunosuppression associated with lymphodepleting chemotherapy, CRS and immune effector cell acute neurotoxicity syndrome (ICANS), as detailed in Supplementary Methods 4. Neurotoxicity that was distinct from ICANS and attributable to local tumour inflammation was designated tumour inflammation-associated neurotoxicity (TIAN)[24,28] and graded using NCI Common Terminology Criteria for Adverse Events v.5.0. For mitigation of risks associated with TIAN, we implemented a toxicity-monitoring and toxicity-management algorithm that (1) placed Ommaya catheters or similar devices for intracranial pressure monitoring and potential treatment of hydrocephalus in all patients; (2) incorporated sequential neurological examinations and scheduled and symptom-prompted intracerebral pressure monitoring; (3) prescribed measures to lower elevated intracranial pressure (positioning, hypertonic saline (3%), cerebrospinal fluid (CSF) removal) in patients with documented elevated intracranial pressure; and (4) administered anakinra and corticosteroids to patients with significant neurological symptoms (detailed in Supplementary Methods 4).

## Demographics

Enrolment began in June 2020 and the data cut-off was 1 December 2023. A consort diagram is shown in Extended Data Fig. 2. Characteristics of the 13 enroled patients are shown in Table 1. Median age was 15 years (range 4–30 years), and seven patients were female; ten patients had DIPG and three had sDMG. The H3K27M mutation was demonstrated in biopsies from all tumours by either immunohistochemistry or DNA sequencing. Median time from diagnosis to enrolment was 5.0 months (range 3.9–11.6 months). Eight patients had disease progression or pseudoprogression on MRI imaging at enrolment, and five did not have documented progression at enrolment. Two patients (nos. 002 and 011) were removed from the study before treatment due to rapid tumour progression and a decline in performance status rendering them ineligible for protocol-directed therapy. Patient no. 002 was treated on a single-patient-compassionate IND, with outcomes for this patient reported previously[2].

## Table 1 | Patient demographics and GD2-CAR T cell infusions received

| UPN | Age (years)/sex | Disease | Histone mutation | Other known mutations and molecular aberrations | At enrolment | | | IV | ICV | |
| --- | --- | --- | --- | --- | --- | --- | --- | --- | --- | --- |
| | | | | | Time since diagnosis (months) | Time since end of XRT (months) | Progression at time of first treatment | Dose (×10⁶ kg⁻¹) | Dose (×10⁶) | No. of doses |
| 001 'DIPG Pt 1'[a] | 14/F | DIPG | H3.3H27M | TP53, KIT amplification, MET amplification, NBN rearrangement | 9.9 | 8.0 | Yes | 1 | NA | 0 |
| 003 'DIPG Pt 2'[a] | 21/M | DIPG | H3.3H27M | TP53, ATRX, RB1 | 15.5 | 12.5 | Yes | 1 | 30 | 4 |
| 004 'DIPG Pt 3'[a] | 5/F | DIPG | H3.3H27M | TP53, DAXX rearrangement | 13.3 | 11.5 | No | 1 | 30 | 2 |
| 005 | 11/F | DIPG | H3.3H27M | NF1, PIK3CA | 4.5 | 1.6 | No | 2 | 30 | 4 |
| 006 | 30/F | sDMG | H3K27M[b] | From cell-free tumour DNA in CSF: TP53, BRCA1, NBN, NF1, MGA rearrangement, CASP8 rearrangement, RAF1 rearrangement | 5.1 | 1.5 | Yes | 2 | 30 | 8 |
| 007 | 4/F | DIPG | H3.3K27M | KDM5A, PTEN | 4.1 | 2.0 | No | 2 | 10, 30, 50 | 16 |
| 008 | 15/M | DIPG | H3K27M[b] | TP53 | 4.6 | 2.0 | Yes | 2 | 30 | 1 |
| 009 | 27/M | sDMG | H3K27M[b] | Patchy ATRX expression | 5.7 | 1.6 | Yes | 2 | 10, 30 | 11 |
| 010 | 17/M | DIPG | H3.3K27M | NF1 | 4.1 | 1.5 | Yes | 2 | 30, 50 | 15 |
| 012 | 12/F | DIPG | H3.3K27M | TP53, FANCM, MYD88, TSC2 | 3.9 | 1.0 | No | 2 | 30 | 1 |
| 013 | 7/M | DIPG | H3.3K27M | TP53 | 4.9 | 2.0 | Yes | 2 | | 0 |

[a]Patient nos. 001, 003 and 004 were previously reported as 'DIPG Patient nos. 1, 2 and 3', respectively, in ref. 2. Note that patient nos. 002 and 011 became ineligible following enrolment and before infusion; patient no. 002 was treated on a single-patient-compassionate IND application and was reported as 'spinal cord DMG patient #1' in ref. 2. [b]As determined by immunohistochemistry. NA, not applicable; UPN, unique patient number; XRT, radiation therapy.

## Cell manufacture and characterization

For the 13 enroled patients, 20 GD2-CART products were successfully manufactured (Extended Data Fig. 1), including seven CAR T cell products remanufactured to support multiple repeated ICV infusions. The product for patient no. 011 was not characterized, because this patient developed rapid tumour progression and died before CAR T cell infusion. Mean manufacturing duration was 7 days. The median time from enrolment to IV GD2-CART infusion was 22.9 days (range 15–38 days). Product functional assessment demonstrated GD2-specific reactivity (Extended Data Fig. 3a,b). Products demonstrated a mean fold T cell expansion of 21.97 (range 12.16–28.06), mean percentage viability of 93.51 (range 86.8–96.9), mean transduction efficiency of 57.27 (range 21.63–83.80) (Supplementary Table 1) and mean vector copy number per CAR⁺ cell of 3.17 (range 0.84–7.20) (Extended Data Fig. 3c,d). Detailed GD2-CART phenotypic composition of 19 products showed a predominance of central memory T cells (Extended Data Fig. 1c–f).

## Treatment received and toxicity

Of 11 patients who received one IV GD2-CART infusion on study, 9 experienced clinical benefit, imaging benefit on MRI or both and received additional ICV infusions, as noted in Table 1 (median of four infusions, range 1–17), administered over 1.0–29.4 months at the time of data cut-off. Incidence and grade of CRS, ICANS and TIAN following each IV and ICV infusion are shown in Fig. 1a, Table 2 and Supplementary Tables 2 and 3. Following IV GD2-CART, all patients experienced CRS, with one of three patients on DL1 experiencing grade 2 CRS, six of eight patients on DL2 experiencing grade 2 or higher CRS and three patients on DL2 experiencing DLT attributed to grade 4 CRS (Table 2). CRS was similar in nature to that observed in other CAR T cell trials[16]. The three instances of grade 4 CRS were characterized by hypotension requiring multiple pressors (one patient), pulmonary oedema and respiratory failure requiring either bilevel positive airway pressure/continuous positive airway pressure (one patient) or intubation (one patient). CRS was managed according to established guidelines using tocilizumab,

anakinra, corticosteroids (dexamethasone and methylprednisolone), fluid resuscitation and supportive care. Following IV GD2-CART, we observed ICANS in one of three patients at DL1 (grade 2) and in four of eight patients at DL2 (n = 1 grade 3, n = 3 grade 1). Based on these results, we identified DL2 (3 × 10⁶ GD2-CART per kg) as exceeding the maximally tolerated dose for IV administration in patients with DIPG/DMG.

Among 62 ICV infusions, no DLTs occurred. Forty-one ICV infusions (66%) were associated with no CRS. Among 19 ICV infusions associated with CRS, most were low grade (n = 16 grade 1; n = 4 infusions in two patients grade 2; n = 1 grade 3 in the context of urosepsis in a patient with sDMG) (Table 2). No ICANS was observed following ICV infusions. One patient (no. 005) experienced a grade 3 sensory neuropathy (Supplementary Table 4) in a stocking-glove distribution following the first infusion, for which vitamin B₆ toxicity was considered a contributory factor; this resolved following vitamin B₆ supplement cessation. Of note, we did not observe any cases of transient painful peripheral neuropathy commonly associated with anti-GD2 antibody therapy[29]. As previously reported[2], one patient (no. 003) experienced a grade 5 intratumoural haemorrhage (Supplementary Table 4) in a region of known intratumoural vascular anomaly. Intratumoural haemorrhages occur relatively frequently in DIPG; risk increases with time from diagnosis, and symptomatic intratumoural haemorrhages occur in around 20% of patients with DIPG by 12 months from diagnosis[30].

We observed TIAN in 91% of patients following IV infusion and in 100% of patients following the first ICV infusion; TIAN grade typically diminished with subsequent infusions (Fig. 1a). Forty-four ICV infusions (71%) were associated with TIAN. Nine of nine (100%) patients with DIPG developed TIAN following IV GD2-CART (n = 2 grade 4, n = 3 grade 3 and n = 4 grade 2), and 67% of patients with DIPG developed TIAN following ICV GD2-CAR T cell infusions (n = 29 of 43 infusions with TIAN: grade 4, n = 1; grade 3, n = 3; grade 2, n = 9; grade 1, n = 16). TIAN reversed in all patients following treatment according to our toxicity management algorithm, and no patient experienced a DLT due to TIAN. In one patient with sDMG, high-grade communicating hydrocephalus was observed during peak tumour inflammation. The agent available

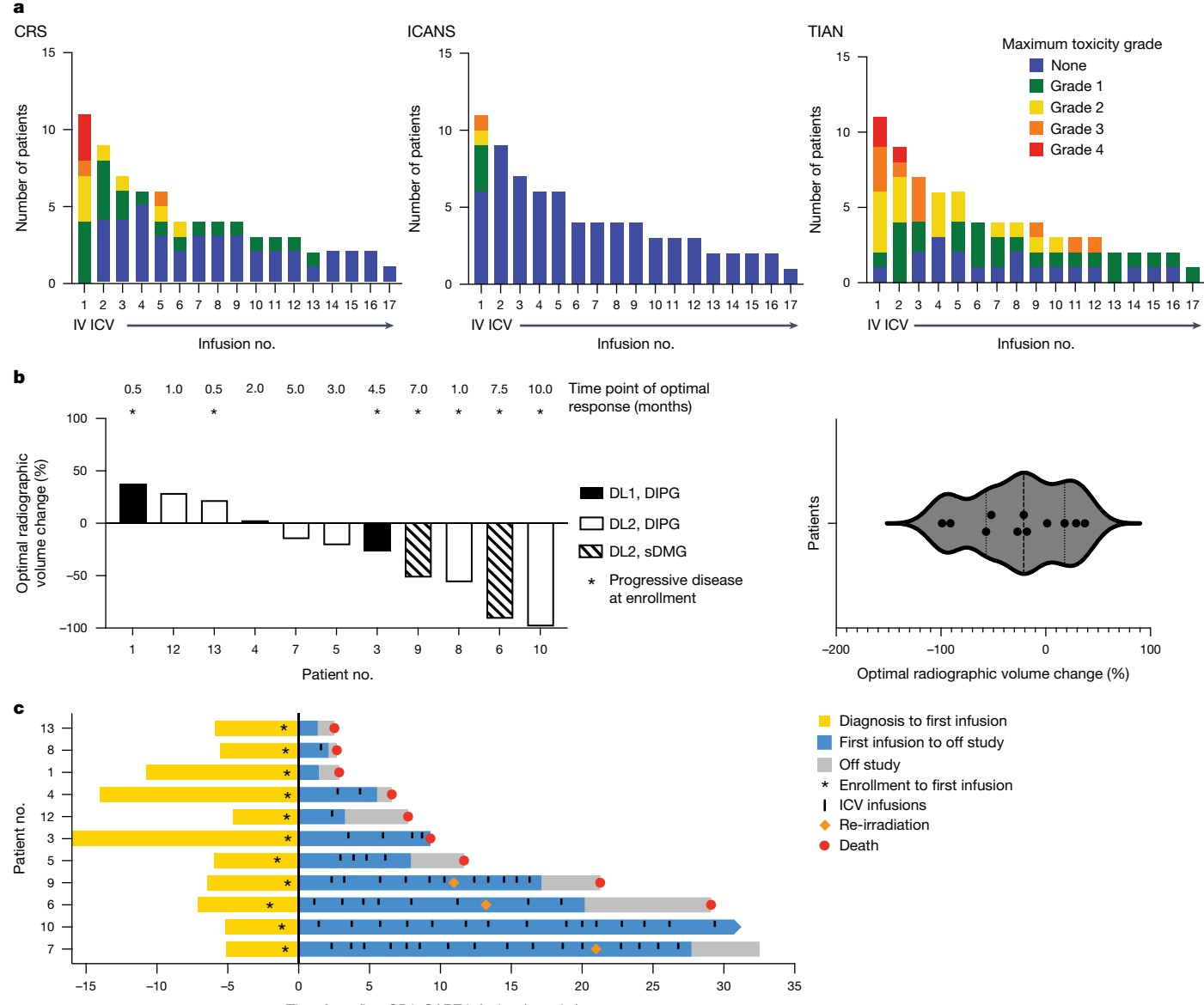

**Fig. 1 | Toxicity and response measures. a**, Number of patients experiencing CRS (left), ICANS (middle) and TIAN (right) following each infusion. Grade of maximal toxicity is indicated by colour in legend. Infusion no. 1 was administered intravenously, infusions 2–17 intracerebroventricularly. **b**, Left, waterfall plot depicting best volumetric change in tumour volume following GD2-CART therapy from baseline measured prior to the first infusion. Asterisks (*) indicate those patients with documented disease progression at the time of first GD2-CAR T cell infusion. The time point at which best radiographic change was measured is noted above the waterfall plot for each patient. Black, patients with DIPG treated at DL1 (*n* = 3 patients); white, patients with DIPG treated at DL2 (*n* = 8 patients); hatched markings, patients with sDMG treated at DL2. Right, violin plot of best volumetric change in tumour volume from baseline, illustrating the normal (Gaussian) distribution of responses; see also Extended Data Fig. 4a for Q–Q plot demonstrating normal distribution of these data.

Each point represents one patient (*n* = 11 patients). **c**, Swimmer plot depicting patient survival. Each bar represents the time from diagnosis to first treatment (yellow), time on trial (blue), time until death (red dot) or data cut-off for individual patients (*n* = 11 patients). Patients remained on trial (blue) until disease progression that was unresponsive to CAR T cell therapy; time elapsing between removal from study for disease progression and death is depicted in grey. Vertical marks indicate each ICV infusion, asterisks indicate time of trial enrolment; first treatment indicated on the *y* axis at time 0. Pause in infusions for focal therapy was allowed per protocol, and three patients (nos. 006, 007 and 009) received re-irradiation, as indicated by orange diamonds. Imaging and clinical benefit in patient nos. 003 and 004 were previously reported[4]; note that, in the previous report[4], our patient no. 001 was described as DIPG Patient 1, patient no. 003 as DIPG Patient 2 and patient no. 004 as DIPG Patient 3.

for ablation of CAR T cells by the induction of caspase-9, AP1903, was not administered to any patients treated on arm A.

## Response

Experience with the first three patients enroled suggested a hypothesized risk of progression approximately 2–3 months following IV infusion[2], including patient no. 003, who showed an impressive

response to IV infusion, followed by progression at approximately day 70, with response to subsequent ICV infusion. Based on this experience, beginning with patient no. 004, the protocol was amended to allow patients with clinical or imaging benefit following IV infusion to receive multiple sequential ICV infusions. Patients were eligible to receive ICV infusions every 1–3 months, providing they had experienced stable disease or clinical or imaging benefit with the previous infusion.

## Table 2 | Maximum toxicity grades following GD2-CAR T therapy

| | IV DL1 n=3 infusions | IV DL2 n=8 infusions | ICV n=62 infusions |
|---|---|---|---|
| **CRS** | n=3 (100%) | n=8 (100%) | n=21 (33.9%) |
| Grade 1 | 2 (66.7%) | 2 (25%) | 16 (76.2%) |
| Grade 2 | 1 (33.3%) | 2 (25%) | 4 (19%) |
| Grade 3 | 0 (0%) | 1 (12.5%) | 1 (4.8%) |
| Grade 4 | 0 (0%) | 3 (37.5%) | 0 (0%) |
| **ICANS** | n=1 (33.3%) | n=4 (50%) | n=0 (0%) |
| Grade 1 | 0 (0%) | 3 (75%) | 0 (0%) |
| Grade 2 | 1 (100%) | 0 (0%) | 0 (0%) |
| Grade 3 | 0 (0%) | 1 (25%) | 0 (0%) |
| Grade 4 | 0 (0%) | 0 (0%) | 0 (0%) |
| **TIAN** | n=3 (100%) | n=7 (87.5%) | n=44 (71%) |
| Grade 1 | 0 (0%) | 1 (14.3%) | 7 (16%) |
| Grade 2 | 2 (66.7%) | 2 (28.6%) | 12 (27%) |
| Grade 3 | 0 (0%) | 3 (42.9%) | 7 (16%) |
| Grade 4 | 1 (33.3%) | 1 (14.3%) | 1 (2.3%) |

Data in the table represent number of infusions associated with a given toxicity at the indicated grade. The percentages next to the number of events represent the percentage of infusions associated with a given toxicity for each toxicity class (CRS, ICANS, TIAN). For each toxicity class the infusions associated with toxicity are distributed by grade.

Several patients exhibited reduction in tumour size following GD2-CART, with the best volumetric change in tumour size for each patient shown in Fig. 1b. In this patient cohort, this 'best response' change in tumour volume data fits a Gaussian distribution by the Shapiro–Wilk test of normality, with no outliers (Fig. 1b and Extended Data Fig. 4a). Overall survival from time of diagnosis to trial enrolment, and from first GD2-CART administration to time off trial, death or data cut-off, is shown in Fig. 1c. Patient no. 010 (DIPG), who was enroled with evidence of progression or possible pseudoprogression by imaging following completion of upfront radiotherapy, demonstrated a continuing reduction in tumour volume to the point of complete response within months following the first GD2-CAR T cell infusion. This complete response was sustained at 30 months following enrolment, the time of data cut-off, and is ongoing (Figs. 1b and 2a,b). Patient no. 006 (sDMG), who was in clear clinical and radiographic tumour progression at the time of first treatment, demonstrated 91% reduction in tumour volume by 7 months following the first GD2-CAR T cell infusion (Figs. 1b and 3a,b). Changes in tumour volume over time for additional patients are shown in Extended Data Fig. 4b.

Clinical improvements, as measured by changes in clinical improvement score (CIS), were also observed (Supplementary Table 5 and Extended Data Fig. 9). In general, clinical improvement coincided with tumour response on MRI (Figs. 2c and 3c). For example, before the first infusion, patient no. 006 (sDMG) had severe paraplegia, neuropathic pain and bowel and bladder dysfunction. At the time of her best overall response (over 90% tumour reduction), she experienced intact bowel and bladder continence, markedly improved pain, and improved lower extremity function enabling ambulation with a cane. Similarly, patient no. 010 (DIPG) had sensory and motor deficits requiring assistance with gait at enrolment (used a wheelchair for longer distances) and experienced a complete response by imaging accompanied by significant clinical improvements, including improved left-sided hearing, improved hemifacial, hemibody and taste sensation, improved motor coordination and improved gait to the point of independent ambulation. For some patients, however, the relationship between imaging findings on MRI and clinical improvement was not direct, with patients no. 004 and no. 007 demonstrating sustained clinical improvement without substantial changes in overall tumour volume by MRI, and patient no. 012 exhibiting early clinical improvement but overall increase in tumour volume (Fig. 1b, Extended Data Figs. 4b and 9 and Supplementary Table 5).

Median overall survival for patients treated on arm A was 20.6 months from diagnosis, with two patients with DIPG alive and followed subsequent to data cut-off (patient no. 007, length of follow-up 33 months; and patient no. 010, length of follow-up 30 months). For patients with DIPG, median overall survival was 17.6 months and for sDMG was 31.96 months (Extended Data Fig. 4c); comparison with historical controls was not possible in this study due to the highly selected nature of the trial cohort.

## Correlative findings

Peripheral blood demonstrated GD2-CART expansion following IV infusion at levels similar to those seen in other active CAR T cell trials[31–33], which persisted during ICV infusions (Figs. 2c and 3c and Extended Data Fig. 5a) but decreased over time. GD2-CART expansion was also evident in CSF following multiple repeated infusions (Extended Data Fig. 5b). Patient nos. 006 (Fig. 3c) and 009 (Extended Data Figs. 4a and 5a) experienced loss of detectable GD2-CART in the peripheral blood, as measured by PCR, temporally correlated with clinical and/or imaging progression. Increased cytokine/chemokine levels were present in peripheral blood following IV GD2-CART, whereas cytokine/chemokine levels were more marked in CSF following ICV GD2-CART (Extended Data Fig. 6), consistent with our previous report of DL1 patients[2]. Sequential ICV GD2-CART infusions induced repeated, transient elevations in CSF cytokine/chemokine levels (Figs. 2d and 3d and Extended Data Fig. 6).

Grade 2 and greater CRS correlated with increased plasma levels of the myeloid cell chemokine MCP1 (also called CCL2), and a trend towards increased levels of the T cell cytokine IL-2 following IV GD2-CART administration (Extended Data Fig. 7a). Grade 2 and greater TIAN correlated with increased CSF levels of IL-10 and MCP1, and a trend towards increased levels of TNF following ICV GD2-CART (Extended Data Fig. 7b). For examination of the correlates of response, we assigned patients to 'responder' and 'non-responder' groups based on their best tumour volume changes (Fig. 1b), with three patients defined as non-responders (overall increase in tumour size) and eight defined as responders (decrease in tumour size or overall stable tumour size and improvement in neurological function). Using this stratification, we found no differences in the CD4:CD8 ratio of the CAR T cell product between responders and non-responders (Extended Data Fig. 7c), but responders exhibited higher levels of IL-2 in plasma following IV GD2-CART administration (Extended Data Fig. 7d), potentially reflecting increased T cell activation in these patients. In individual patients, progression was temporally associated with a decrease in plasma IP-10 or CXCL10 (Extended Data Fig. 7e). CSF levels of immunosuppressive TGF-beta factors increased following GD2-CART administrations in some patients, and higher TGF-beta1 levels early following GD2-CART infusion trended with progression of disease (Extended Data Fig. 8a–c). Notably, TGF-beta levels in CSF were relatively low in patient no. 010, who had a complete response, throughout his course (Extended Data Fig. 8a,b).

Cell-free tumour DNA (ctDNA) in CSF, as measured by ddPCR, demonstrated decreased H3K27M mutations per millilitre CSF in some patients associated with tumour regression, and increased levels at times of peak inflammation or when patients experienced clinical and/or imaging progression (Figs. 2c and 3c and Extended Data Fig. 5a). Taken together, these data demonstrate evidence for sustained expansion and persistence of GD2-CART in the blood of patients with DIPG and sDMG, associated with clinical and biological evidence of antitumour activity, and begin to identify potential contributors to, or correlates of, loss of durable response in patients.

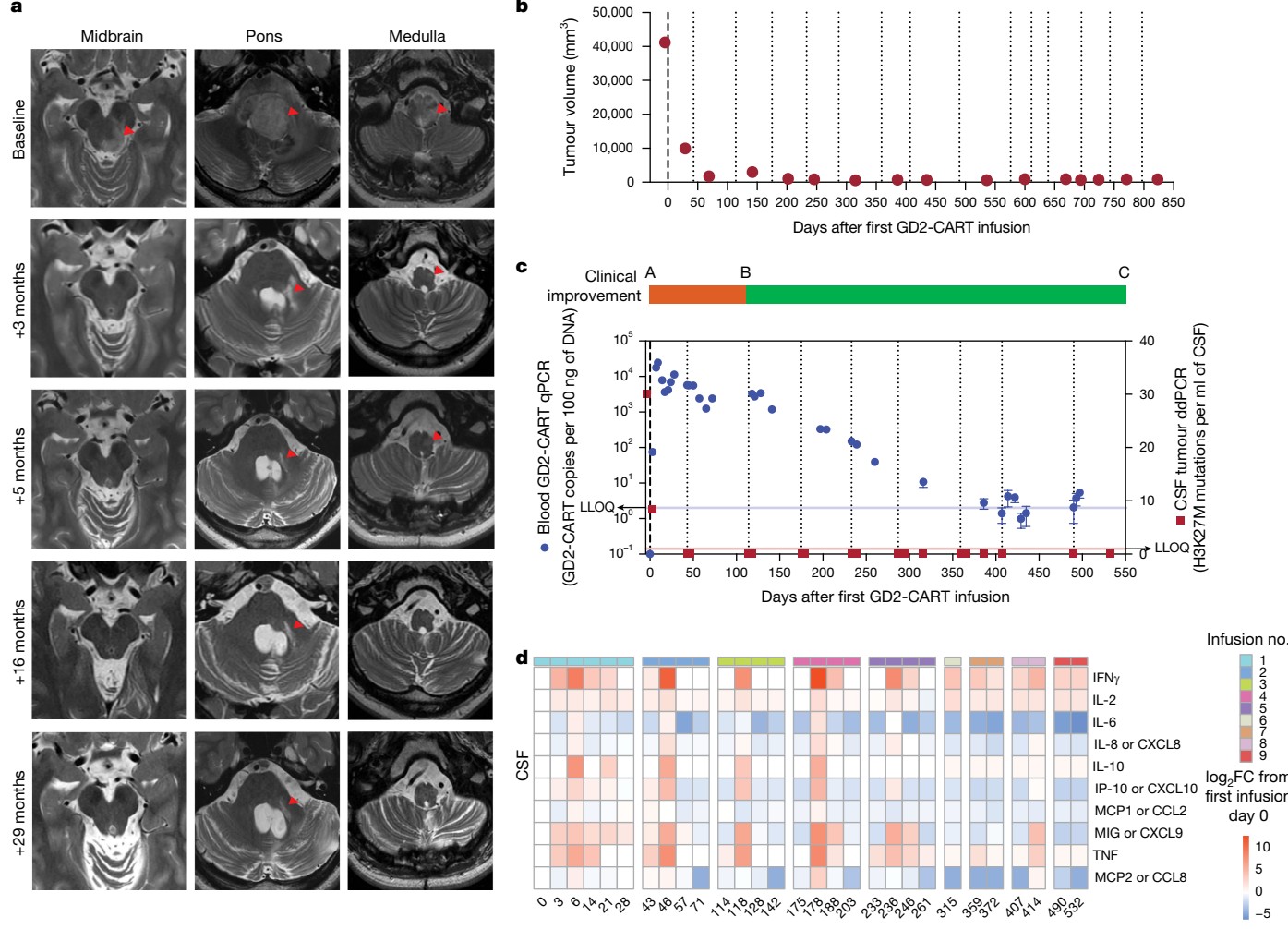

**Fig. 2 | Therapeutic response and correlative findings for patient no. 010.**
**a**, T2-weighted, axial MRI images of tumour at midbrain, pons and medulla levels at baseline and at 3, 5, 16 and 29 months following first infusion. Red arrowheads indicate T2 signal abnormality. At baseline, extensive tumour involving the midbrain (left more than right), pons and medulla is evident, with mass obscuring the fourth ventricle at baseline resolving by 3 months. At 3 months, return of CSF around the brainstem is evident as the size of the brainstem normalizes, and T2 signal normalizes throughout much of the brainstem. At 3 months, T2 signal abnormality, probably represening tumour, remains around the area of the biopsy tract in the pons (red arrowhead) and resolves by 5 months. At 5, 16 and 29 months, the arrowheads indicate stable T2 signal abnormality of the biopsy tract and stable area of subtle T2 hyperintensity of unclear significance in the medulla (as standard for axial MRI images, the patient's left is the reader's right). **b**, Tumour volume as a function of days following first GD2-CAR T infusion. **c**, Overlay of clinical change, CAR T cell

persistence and tumour cell-free DNA. Clinical improvement is depicted as a bar at the top of the panel: red, clinical worsening from baseline; green, clinical improvement from baseline. CAR T cell persistence in blood, as measured by CAR construct quantitative PCR (qPCR, blue data points, left $y$ axis). Cell-free tumour DNA (H3K27M) in CSF by digital droplet PCR (ddPCR, red data points, right $y$ axis). GD2-CART infusions indicated as dotted vertical lines. Each qPCR data point represents the mean of three technical replicates; each ddPCR data point represents the mean of four technical replicates; error bars represent s.e.m. **d**, CSF cytokine levels following each infusion, expressed as $\log_2$ fold change (FC) from day 0 before the first infusion. Time points are days from first infusion (first infusion administered intravenously, subsequent infusions administered intracerebroventricularly); infusions are separated by vertical white spacing; infusion number indicated by coloured bar above each heatmap; each square of the heat map represents one biological replicate. LLOQ, lower level of quantitation.

## Discussion

Recent advances in fundamental understanding of DMG have begun to translate to clinical advances[21–23,34–36]. The present study establishes GD2-CAR T cell therapy as a promising modality for a historically lethal CNS cancer. The rate of clinical improvements and tumour regressions on MRI imaging, including a complete response by RANO 2.0 criteria sustained for over 30 months and throughout the duration of this report, is reason for cautious optimism, both for DIPG/sDMG therapy and, more broadly, for CAR T cell therapy of solid tumours. Therapeutic activity was clearly demonstrated in several patients, based on tumour regression by imaging and improvements in neurological symptoms.

Whether GD2-CAR T cell therapy favourably impacts overall survival is difficult to discern, given that the eligibility criteria excluded patients with lower performance status, the need for steroid therapy, bulky thalamic or cerebellar tumour involvement and other factors. Moreover, multiple patients demonstrated MAPK pathway and ATRX or DAXX mutations, which can be associated with relatively longer survival[37] and more localized disease[38]. The effects of GD2-CAR T cell therapy on overall survival will need to be assessed in future Phase II trials.

The toxicities observed here were generally anticipated, but with some notable observations. Cytokine release syndrome was dose limiting following IV infusion of GD2-CART for DIPG/DMG. By contrast, CRS did not commonly complicate ICV infusions. Given that the antitumour

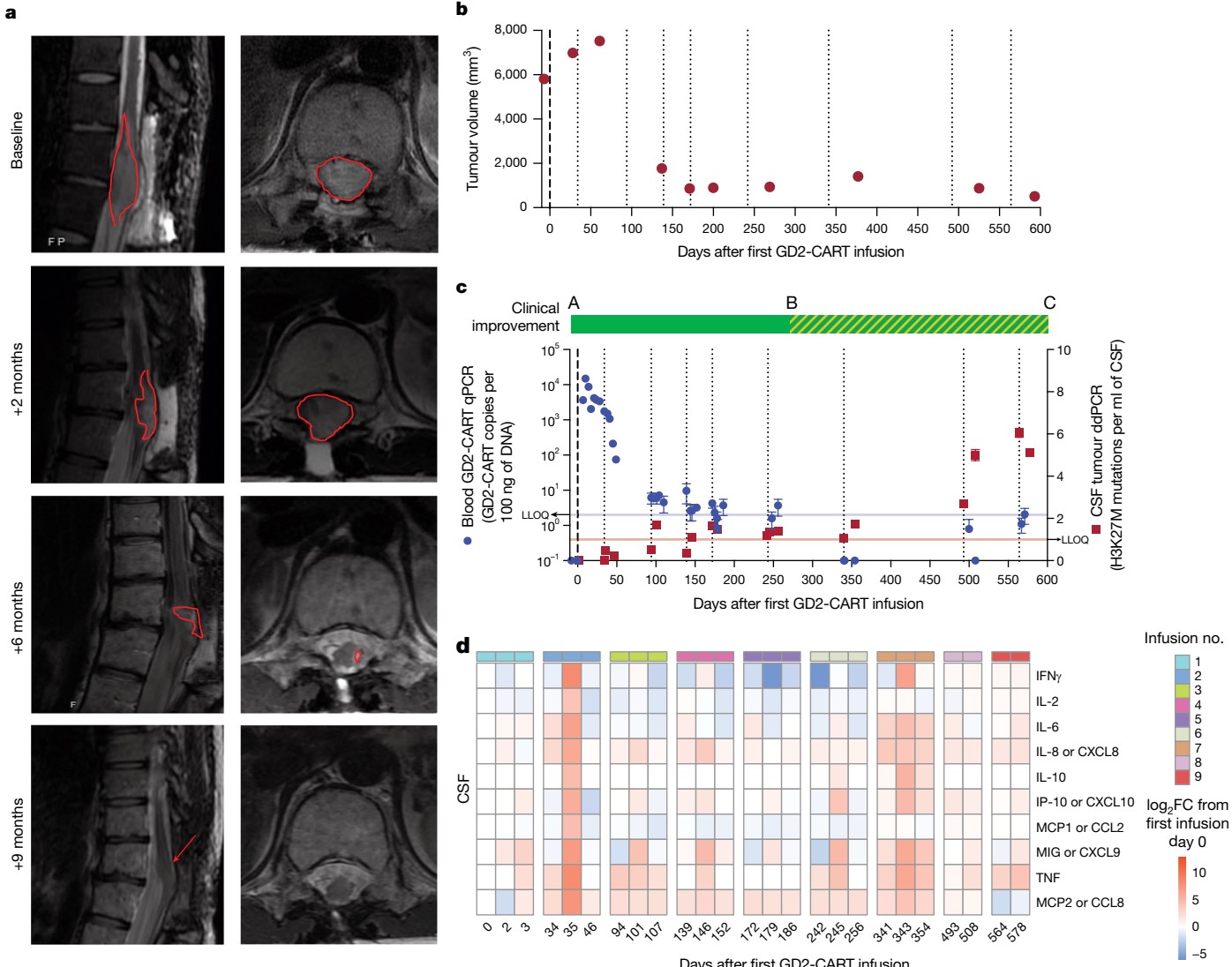

**Fig. 3 | Therapeutic response and correlative findings for patient no. 006.**
**a**, T2-weighted, sagittal (left) and axial (right) MRI images of spinal cord DMG at baseline and at 2, 6 and 9 months following first infusion. Red outline indicates T2 signal abnormality (tumour). At baseline, tumour centred at T11/12 spinal level diffusely involves the spinal cord and expands the cord to fill the entire spinal canal, with no CSF visualized around cord. Tumour infiltration of the cord progressively improves until it is of normal calibre and T2 signal abnormality is minimal (red arrow). **b**, Tumour volume as a function of time (days) following first GD2-CART infusion. **c**, Overlay of clinical change, CAR T cell persistence and tumour cell-free DNA. Clinical improvement is depicted as a bar at the top of the panel, with solid green indicating clinical improvement from baseline and hatched green indicating improvement from pretreatment baseline but worsened from peak improvement. CAR T cell persistence in blood, as measured by qPCR of the CAR construct, is denoted by blue circles (left y axis); cell-free

tumour DNA (H3K27M) in CSF, as measured by ddPCR, is indicated by red squares (right y axis). Each qPCR data point represents the mean of three technical replicates; each ddPCR data point represents the mean of four technical replicates; error bars represent s.e.m. GD2-CART infusions indicated as dotted vertical lines. Note that, as CAR T cells become undetectable in blood by qPCR, cell-free tumour DNA elevates; this inflection point correlates with the beginning of disease progression. **d**, CSF cytokine levels following each infusion, expressed as log$_2$ fold change from day 0 before the first infusion. Time points represent days from first infusion (first infusion administered intravenously, subsequent infusions administered intracerebroventricularly); infusions separated by vertical white spacing; infusion number indicated by coloured bar above each heatmap; each square of the heatmap represents one biological replicate.

immune response occurs within the CNS in both contexts, this marked difference in systemic cytokine responses leading to CRS following IV administration is somewhat unexpected, and underscores the unique features of immune cell trafficking into and out of the CNS only recently coming to light[39,40]. Unexpectedly, ICANS was observed only following IV administration, but not with ICV infusions, despite the higher levels of cytokines/chemokines evident in CSF following ICV GD2-CART infusions. Whereas the rate of intratumoural haemorrhage observed in this cohort (1 of 11) is in line with the natural history of this disease[30], future study is needed to determine whether CAR T cell therapy alters the risk of intratumoural bleeding. Inflammation of neural structures

can cause oedema with consequent mechanical complications such as obstruction of CSF flow, and immune signalling can also affect neuronal function. Varying grades of TIAN[28] occurred commonly, as would be expected with inflammation of tumours located in structurally constrained and functionally eloquent neuroanatomical areas such as the brainstem and spinal cord. Concordantly, tumours located in the lower brainstem were associated with more severe TIAN. TIAN was transient and manageable with intensive support. In general, earlier infusions and larger tumours were associated with greater toxicity. The correlation of more severe TIAN with higher levels of cytokines/chemokines related to myeloid response, such as MCP1 or CCL2, raises the prospect that

myeloid cells may contribute to neurological symptoms induced by CAR T cell therapy for CNS tumours, a hypothesis that requires testing in future studies. The observed decrease in TIAN severity with repeated infusions is notable and could be due to decreased tumour burden with repeated infusions, increased immunomodulatory or immunosuppressive mechanisms in the tumour microenvironment, or both.

Given the promising results noted in arm A with repeated ICV administration of GD2-CAR T cells, but with dose-limiting CRS following IV administration at high dose levels, we have now initiated arms B and C to test ICV-only administration with and without lymphodepleting chemotherapy. Given the burden of monthly infusions on the patient, we have also included formalized assessments of quality of life and patient-reported outcomes in arms B and C. Ongoing and future work will define the optimal route of administration, the role of lymphodepleting chemotherapy and potential combination strategies for optimization of this promising therapy to achieve more complete and durable responses for children and adults with H3K27M-mutated diffuse midline gliomas.

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

[1]Department of Neurology and Neurological Sciences, Stanford University, Stanford, CA, USA. [2]Division of Pediatric Hematology/Oncology/Stem Cell Transplant and Regenerative Medicine, Department of Pediatrics, Stanford University, Stanford, CA, USA. [3]Stanford Center for Cancer Cell Therapy, Stanford Cancer Institute, Stanford University, Stanford, CA, USA. [4]Department of Neurosurgery, Stanford University, Stanford, CA, USA. [5]Department of Pathology, Stanford University, Stanford, CA, USA. [6]Howard Hughes Medical Institute, Stanford University, Stanford, CA, USA. [7]Division of Neuroradiology, Department of Radiology, Stanford University, Stanford, CA, USA. [8]Cellular Therapy Facility, Stanford Health Care, Palo Alto, CA, USA. [9]Department of Pediatrics, Division of Pediatric Critical Care Medicine, Stanford University, Stanford, CA, US. [10]Department of Neurosurgery, Johns Hopkins School of Medicine, Baltimore, MD, USA. [11]Parker Institute for Cancer Immunotherapy, San Francisco, CA, USA. [12]Division of Stem Cell Transplantation and Cell Therapy, Department of Medicine, Stanford University, Stanford, CA, USA. ✉e-mail: mmonje@stanford.edu; ramakrs@stanford.edu; cmackall@stanford.edu

## Methods

### Human subjects

The Stanford Institutional Review Board approved this trial, which was registered with ClinicalTrials.gov (NCT04196413). Informed patient or parent consent and child assent were obtained. Included in consent is permission to share the (deidentified) results of the trial in scientific publications or presentations. Permission to publish images, photographs and videos was also obtained.

### Response assessment

Clinical response was assessed using the CIS[2] generated by protocol-directed neurological examinations performed by a neuro-oncologist at prescribed times following GD2-CAR T cell administration. The CIS represents a simple quantification of the neurological examination, which added or subtracted one point for each symptom/sign that improved or worsened, respectively, from the patient's preinfusion baseline examination (Supplementary Methods 5). For patients who received corticosteroid therapy for toxicity management, CIS scoring was deferred until at least 7 days following corticosteroid discontinuation. Imaging responses were assessed by MRI scans of the brain and/or spinal cord with and without gadolinium. Because DMGs are diffusely infiltrative of CNS structures and difficult to measure in linear dimensions, volumetric segmentation of tumour corresponding to abnormal T2 signal was performed by a neuroradiologist to measure radiographic change in tumour volume, consistent with RANO 2.0 recommendations[41].

### Statistical analysis

Sample size estimation was based on clinical considerations and the Phase I 3 + 3 design. The targeted DLT rate was 30% or less. All patients who enroled into arm A and received one infusion of GD2-CAR T cells on trial were included in the analysis. Descriptive statistics were used to summarize baseline patient and disease characteristics, toxicity data and correlative and clinical outcomes. The Shapiro–Wilk test of normality was used to assess normal (Gaussian) distribution of tumour volumetric response data. Overall survival was measured from date of diagnosis to the date that event occurred, or censored at the time of data cut-off. Survival probability was estimated using the Kaplan–Meier method[42]. The confidence interval of median survival time was constructed by the method of Brookmeyer–Crowley[43]. All comparisons made in correlative outcomes were exploratory. Mann–Whitney $U$-test was used, with no adjustments made for multiple comparisons. Statistical analyses were conducted using Prism software.

### Correlative studies

Peripheral blood and CSF samples were collected before and following IV and ICV infusions to measure cytokine/chemokine levels in blood and CSF, GD2-CART persistence by qPCR and CAR-FACS and cell-free tumour DNA in CSF, as detailed in the previous report of the first four patients[2] and reiterated below.

**qPCR measurement of in vivo GD2-CAR expansion.** Patient blood samples were processed and mononuclear cells viably cryopreserved. DNA was extracted from whole blood ($2 \times 10^6$–$5 \times 10^6$ peripheral blood mononuclear cells (PBMCs)) using the QIAmp DNA blood Mini Kit (Qiagen, catalogue no. 51306) at baseline and at multiple time points following CAR administration. CAR presence was measured by qPCR using the primer and probe sequences provided below. For the standard curve, a custom Minigene plasmid (IDT) was designed containing a partial GD2.41BB.z sequence and a partial albumin sequence, which served as a control for normalization. The standard curve contained a tenfold serial dilution of plasmid ranging between $5 \times 10^8$ and $5 \times 10^0$ copies. Both plasmid and patient DNA from each time point were run in triplicate, with each reaction containing 5 μl of DNA (50 ng total),

200 nM forward and reverse albumin primers (or 300 nM forward and reverse GD2.41BB.z primers), 150 nM probe suspended in 10 μl of TaqMan Fast Universal PCR Master Mix (2X), no AmpErase UNG (Thermo Fisher Scientific) and either 24.5 μl (albumin) or 22.5 μl (GD2.41BB.z) of TE buffer (Invitrogen, catalogue no. AM9935). The Thermo Fisher Scientific QuantStudio 6 Pro Real-time qPCR Instrument was used for qPCR, with 20 μl per reaction. Quality metrics for all qPCR standard curve results were $R^2 > 0.95$ and efficiency 70–110%.

Albumin results from the plate were normalized to average albumin, then GD2-CAR copy number (copies per 50 ng of DNA), adjusted to albumin and modified to copies per 100 ng of DNA, was calculated by the following equation: copy number (copies per 100 ng of DNA) = 2 × (GD2-CAR copy number × (albumin copy number/average albumin)).

**Real-time flow cytometry assay.** A high-dimensional, immunophenotyping, flow cytometry panel was designed for immune profiling of CAR T cells in real time. PBMCs were isolated from fresh whole blood by gradient centrifugation on ficoll (Ficoll paque Plus, GE Healthcare, Sigma-Aldrich). Between 2 million and 5 million PBMCs were stained with fixable Live/Dead aqua (Invitrogen) amine-reactive viability stain. Cells were preincubated with Fc block (trustain, BioLegend) for 5 min, then stained at room temperature with fluorochrome-conjugated mAb in a 15-colour, 17-parameter staining combination as previously described[2]. Antibody–fluorochrome conjugates used were: anti-CD3-FITC (BioLegend, 0.5 μg of antibody per 1 million cells in 100 μl); anti-CD8-PerCP Cy5.5 (BD Biosciences, 0.5 μg of antibody per 1 million cells in 100 μl); anti-CD45-BV785 (BioLegend, 1 μg of antibody per 1 million cells in 100 μl); anto-CD4-BV711 (BioLegend, 1 μg of antibody per 1 million cells in 100 μl); anti-CD95-BV650 (BioLegend, 0.5 μg of antibody per 1 million cells in 100 μl); CD39-BV605 (BioLegend, 1 μg of antibody per 1 million cells in 100 μl); anti-CD57-BV421 (BioLegend, 1 μg of antibody per 1 million cells in 100 μl); anti-CCR7-BUV805 (BioLegend, 1 μg of antibody per 1 million cells in 100 μl); anti-CD45RA-Alx700 (BioLegend, 0.5 μg of antibody per 1 million cells in 100 μl); anti-GD2-CAR-DyLight650 (custom, clone 1A7, 1 μg of antibody per 1 million cells in 100 μl); anti-CD14-PE-Cy7 (BioLegend, 1 μg of antibody per 1 million cells in 100 μl); anti-CD11b-APC-Cy7 (BioLegend, 1 μg of antibody per 1 million cells in 100 μl); anti-CD33-PE-Dazzle (BioLegend, 5 μl of antibody per 1 million cells in 100 μl); and anti-GD2-PE (BioLegend, 5 μl of antibody per 1 million cells in 100 μl). For determination of the optimal concentration of antibody for staining in the CAR-FACS panel, each fluorochrome-conjugated antibody was titrated to determine the saturating amount of antibody needed to stain a test of 1 million cells. Dose–response curves for each antibody informed the saturating amount of fluorochrome-conjugated antibody required for staining 1 million cells in 100 ml of staining volume.

CAR-tranduced T cells were used as positive control included in daily staining experiments. Immunostained and fixed cells were acquired on an LSR (BD BioSciences) five-laser (blue, 488 nm; violet, 405 nm; ultraviolet, 355 nm; red, 640 nm; green, 532 nm) analyser. A minimum of $10^6$ cells were acquired, unless restricted by the number of cells isolated from 8 ml of whole blood or when acquiring isolated cells from CSF. The assay limit of detection for cells was calculated as 1 in $10^4$ of total acquired PBMCs. Representative gating is shown in Extended Data Fig. 10.

**Luminex cytokines.** Patient blood and CSF samples were collected at predetermined and trigger time points throughout treatment. Samples were spun at 250$g$ for 6 min. Supernatant was frozen at −80 °C until batched for assessment. Cytokine assessment was performed with the Immunoassay Team-Human Immune Monitoring Center at Stanford University. Panels include Luminex–EMD Millipore HIMC H80 (panel 1 is Milliplex HCYTA-60K-PX48; panel 2 is Milliplex HCP2MAG-62K-PX23; panel 3 includes Milliplex HSP1MAG-63K-06 and HADCYMAG-61K-03

(resistin, leptin and hepatocyte growth factor) to generate a nine-plex) and TGF-b (TGFBMAG-64K-03). Kits were purchased from EMD Millipore and used according to the manufacturer's recommendations, with modifications described. The assay set-up followed the recommended protocol. Briefly, samples were diluted threefold for panels 1 and 2 and tenfold for panel 3. Diluted sample (25 µl) was mixed with antibody-linked magnetic beads in a 96-well plate and incubated overnight at 4 °C with shaking. Both cold and room-temperature incubation steps were performed on an orbital shaker at 500–600 rpm. Plates were washed twice with wash buffer in a Biotek ELx405 washer (BioTek Instruments). Following incubation for 1 h at room temperature with biotinylated detection antibody, streptavidin-PE was added for 30 min with shaking. Plates were washed as described above, and PBS added to wells for reading in the Luminex FlexMap3D Instrument with a lower bound of 50 beads per sample per cytokine. Custom Assay Chex control beads (Radix Biosolutions) were purchased and added to all wells. Wells with a bead count under 50 were flagged, and data with a bead count under 20 were excluded. Data are presented as picograms per millilitre, based on standard curves or heat maps of fold change from the baseline time point. All samples were run in technical duplicate.

## Reporting summary

Further information on research design is available in the Nature Portfolio Reporting Summary linked to this article.

## Data availability

Source data are provided with this paper.

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

**Acknowledgements** We thank the patients and their families who participated in this clinical trial. We also thank Bellicum Pharmaceuticals for providing the GD2-CAR vector and AP1903. We thank Stanford HIMC for performing cytokine assays, and the Cellular Therapy Facility at Stanford for their support of this Phase I clinical trial, including that for cell pharmacy. Funding for the clinical trial was provided by grants from the Parker Institute for Cancer Immunotherapy, CureSearch, the National Cancer Institute (no. R01CA263500) and the California Institute for Regenerative Medicine (no. CLIN2-12595). This work was also supported in part by grants from the St. Baldrick's Foundation to the Empowering Immunotherapies for Children's Cancers Cancer team and Alex's Lemonade Stand Foundation. We acknowledge generous funding by the Virginia and D. K. Ludwig Fund for Cancer Research (M.M. and C. Mackall), ChadTough Defeat DIPG Foundation (M.M. and J. Mahdi), Oscar's Kids Foundation (M.M.), Arms Wide Open Childhood Cancer Foundation (M.M.), Tough2gether Foundation (M.M.), Storm the Heavens Foundation (M.M.), the DIPG/DMG Research Funding Alliance (M.M.) and the Chambers-Okamura Endowed Directorship for Pediatric Neuro-Immuno-Oncology (M.M.). R.M. was the Taube Distinguished Scholar for Pediatric Immunotherapy at Stanford University School of Medicine, and was supported by the Be Brooks Brave Fund St. Baldrick's Scholar Award. S.R. was supported by K08CA267057.

**Author contributions** M.M. is principal investigator of the trial and C. Mackall is the IND holder. R.M., C. Mackall and M.M. conceived of the project. S.M., E.E. and C.B. oversaw regulatory affairs. S.M., E.E., M.M., C. Mackall, R.M., S. Partap., C.J.C., J. Mahdi, L.R., T.T.C. and G.G. planned, designed and/or wrote the clinical trial, amendments and treatment protocols. S. Patel., H.C. and S.A.F. performed process development for cellular manufacturing and supervised cellular manufacturing. A.K.B., A.-L.G.P., C. McIntyre, C.F., A.K.D., R.T. and D.D.K. participated in cellular manufacturing and preparation for infusion. R.M., S.R., K.-W.S., L.M.S., R.M.R., V.B., J. Mahdi, A.R., C.B., J. Moon, C.E., S.G., L.R., T.T.C., S. Partap, P.G.F., C.J.C., G.G., L.P., K.L.D., C. Mackall and M.M. participated in patient care. K.W.Y. read MRI images and performed volumetric analysis of MRI scans. C.B., J. Moon, M.K., A.S.L. and M.F. participated in the collection and/or processing of patient samples. V.B., W.D.R. and B.S. performed correlative studies. V.B. designed the ctDNA assay. S.R., S.P.R., S. Prabhu and B.S. analysed correlative studies. C.C. and B.B. provided administrative and programme management support. X.Y. provided statistical contributions. J. Mahdi., S.R., M.M. and C. Mackall prepared figures and wrote the manuscript. All authors reviewed and edited the manuscript. M.M. and C. Mackall supervised all aspects of the work.

**Competing interests** M.M., R.M. and C. Mackall are coinventors on a patent for the use of GD2-CAR T cells in regard to H2K27M gliomas, coordinated through Stanford University. C. Mackall is a coinventor on patents for the use of dasatinib and other small molecules to modulate CAR function and control CAR-associated toxicity, and on several patents related to CAR T cell therapies. C. Mackall holds equity in CARGO Therapeutics, Link Cell Therapies and GBM Newco, which are developing CAR-based therapies, and consults for CARGO, Link, Immatics, Ensoma, GBM NewCo and Red Tree Capital. She receives research funding from Lyell and Tune Therapeutics. R.M. is a cofounder of, and holds equity in, Link Cell Therapies and CARGO Therapeutics, and is a consultant for Lyell Immunopharma, NKarta, Arovella Pharmaceuticals, Innervate Radiopharmaceuticals, Aptorum Group, Gadeta, FATE Therapeutics (Data and Safety Monitoring Board) and Waypoint Bio. S.A.F. holds several patents in the field of cellular immunotherapy. V.B. is an investor and Director at Umoja Biopharma and Arsenal Bio. S.P.R. holds equity in Lyell Immunopharma. M.M. holds equity in MapLight Therapeutics and CARGO Therapeutics. The other authors declare no competing interests.

**Additional information**
**Correspondence and requests for materials** should be addressed to Michelle Monje, Sneha Ramakrishna or Crystal Mackall.

**a**

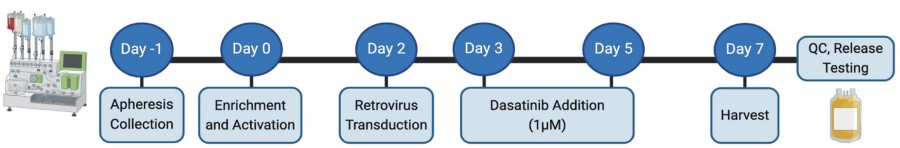

**b**

14G2α

CD8 TM

4-1BB

CD3ζ

iCasp9

**c**

**T Cells**
(CD3+CD56-)

**NKT-like**
(CD3+CD56+CD16+)

**NKs**
(CD3-CD56+CD16+)

**B Cells**
(CD20+)

% of Cells

APH ENR CART

**Monocytes**
(CD14+)

**Neutrophils**
(CD14-CD20-CD16+ high side scatter)

**Eosinophils**
(CD14-CD20-CD16- high side scatter)

% of Cells

APH ENR CART

● Apheresis
● CD4+, CD8+ enriched T Cells
● GD2-CART drug product

**d**

**CD8+ T Cell Population**

% of Total Product

Apheresis Enrichment Drug Product

Manufacturing Timepoint

**CD4+ T Cell Population**

% of Total Product

Apheresis Enrichment Drug Product

Manufacturing Timepoint

**e**

**Memory Marker Expression on CD8+ Cells**

% of CD3+CAR+ Cells

<0.0001

SCM CM EM TEMRA

**Memory Marker Expression on CD4+ Cells**

% of CD3+CAR+ Cells

<0.0001

SCM CM EM TEMRA

**f**

**Exhaustion Markers Expression on CD8+ Cells**

○ CAR-
● CAR+

% of CD3+CAR+ Cells

<0.0001

CD39+ LAG3+ PD-1+ TIM3+

**Exhaustion Markers Expression on CD4+ Cells**

● CAR+
○ CAR-

% of CD3+CAR+ Cells

<0.0001

CD39+ LAG3+ PD-1+ TIM3+

**Extended Data Fig. 1 | See next page for caption.**

**Extended Data Fig. 1 | GD2-CAR T cell manufacturing process and drug product characterization. a**. GD2-CART manufacturing workflow from patient apheresis collection to final drug product harvest across 7 days. **b**. Schematic representing GD2.4-1BB.CD3ζ CAR containing 14G2α binding domain, CD8α hinge and transmembrane (TM) domain, 4-1BB co-stimulatory domain, and a CD3ζ domain. **c**. Flow cytometric immunophenotyping of apheresis, CD4⁺/CD8⁺ enriched cells, and final drug product demonstrates a predominance of CD3 + T cells and depletion of other subsets. **d.** The CD4:CD8 ratio was 1.63 ± 1.19 (SD) at apheresis, and 2.48 ± 0.06 (SD) in the GD2-CART final drug product. **e**. Phenotyping of T-cell subsets shows a predominance of central memory (CM) T cells (CD45RA⁻CCR7⁺), followed by effector memory (EM) cells (CD45RA⁻CCR7⁻), in both CD8⁺ (left) and CD4⁺ (right) subsets in the final cell product. **f.** Exhaustion marker expression on CAR⁺ and CAR⁻ populations reveal no significant differences in CD39, LAG3, and TIM3 expression, while a significantly greater percentage (two-tailed unpaired t-test) of CAR⁺ cells express of PD-1. For all graphs, n = 19 GD2-CART products across 12 patients. Data includes 7 additional GD2-CART products re-manufactured for these patients to enable additional infusions. One point =1 GD2-CART product. Schematics in **a** and **b** created using BioRender (https://biorender.com).

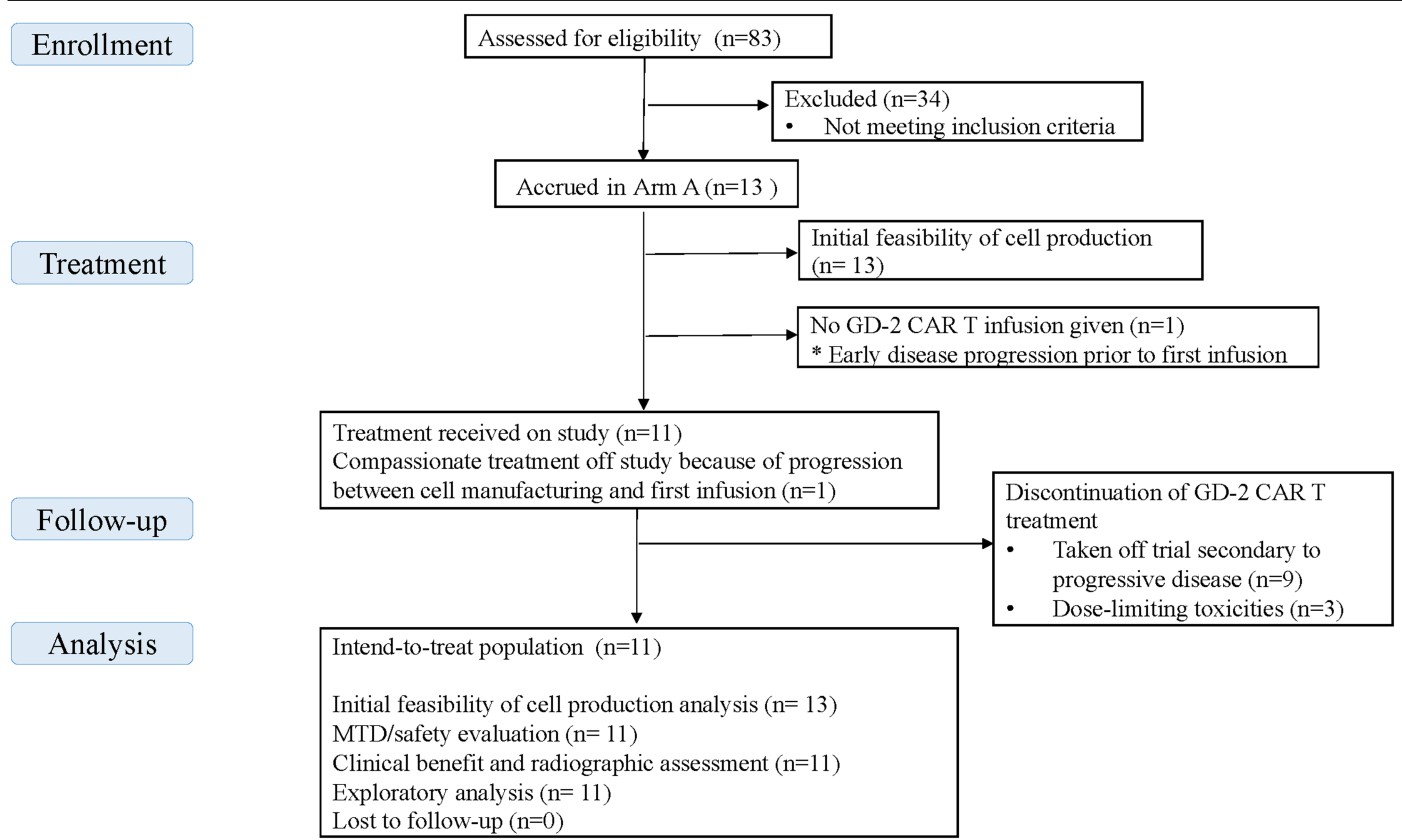

**Extended Data Fig. 2 | CONSORT diagram.** 83 patients were assessed for eligibility. Of these, 34 were not eligible, 13 were enrolled, and 36 patients either chose other trials, chose not to enroll on a clinical trial, or progressed to the point of ineligibility while on the waitlist.

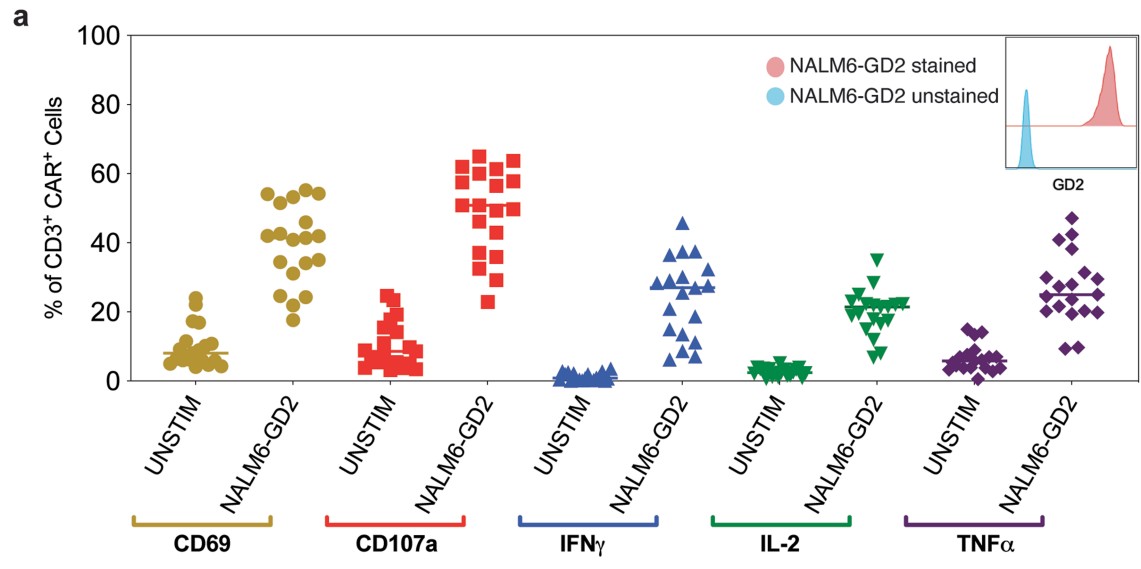

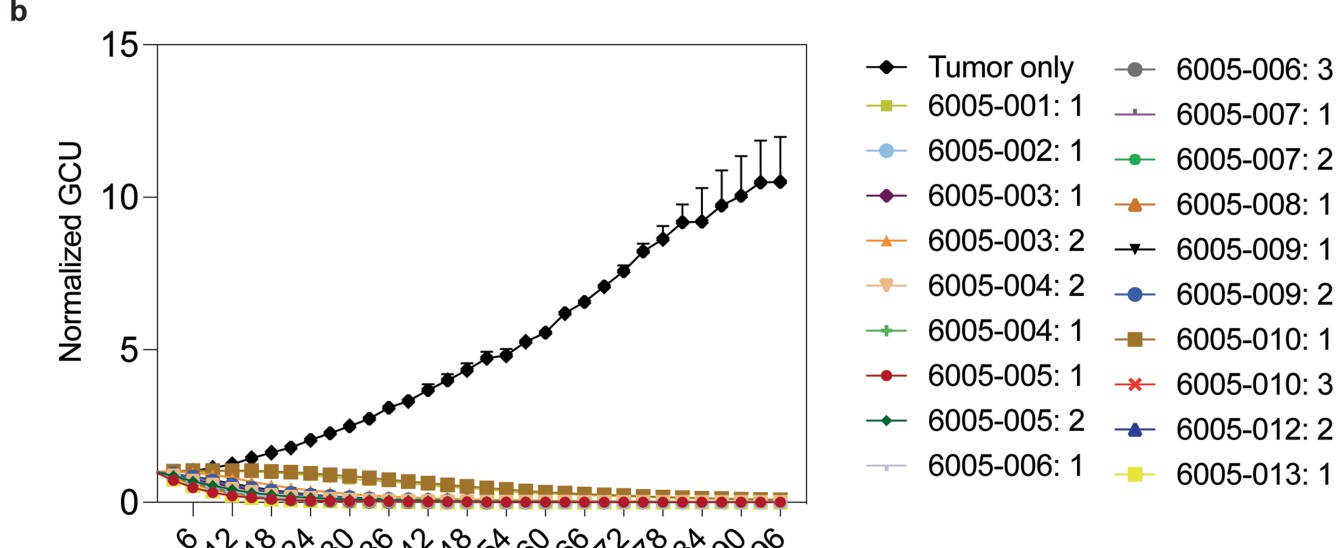

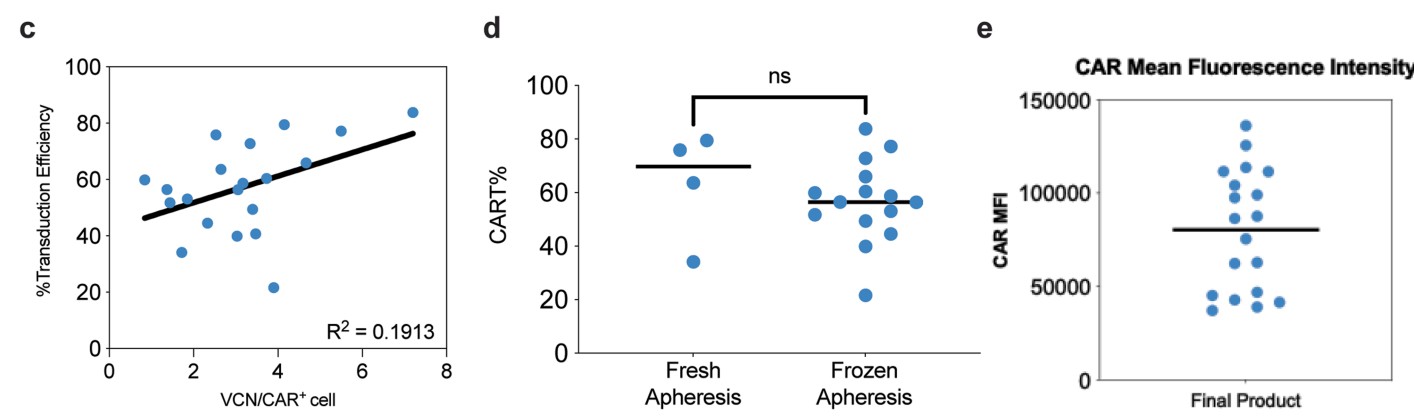

**Extended Data Fig. 3** | See next page for caption.

**Extended Data Fig. 3 | Functional assessment of GD2-CAR T cells in vitro.**
**a**. Inset: GD2 expression on NALM6-GD2 cell line (acute lymphoblastic leukemia cells engineered to express GD2) showing 38,379 and 0.5 molecules per cell, for stained (red) and unstained (blue) populations, respectively. GD2-CAR T cell reactivity to NALM6-GD2 tumor cells measured by intracellular cytokine secretion (ICS) shows upregulation of CD69, CD107a and secretion of interferon-γ (IFNγ), interleukin-2 (IL-2) and tumor necrosis factor-α (TNFα). **b**. Incucyte based cytotoxic activity (E:T 1:1), showed that each patient drug product-controlled tumor cell growth within 96 h of co-culture. Each data point is the mean of three technical replicates; error bars, standard error of the mean. **c**. Transduction efficiency for each patient product plotted as a function of VCN per CAR$^+$ cell ($R^2 = 0.1910$). **d**. No significant difference (ns = p > 0.05, two-tailed unpaired t-test) was observed in GD2-CAR transduction efficiency when manufacturing runs were executed using fresh or frozen patient apheresis material. **e**. CAR mean fluorescence intensity (MFI) values for CAR T cell expression. For all graphs, n = 19 GD2-CART products across 12 patients. Data includes 7 additional products re-manufactured for these patients to enable additional infusions. For a and c, one point = one GD2-CART product. For b, one line = one GD2-CART product.

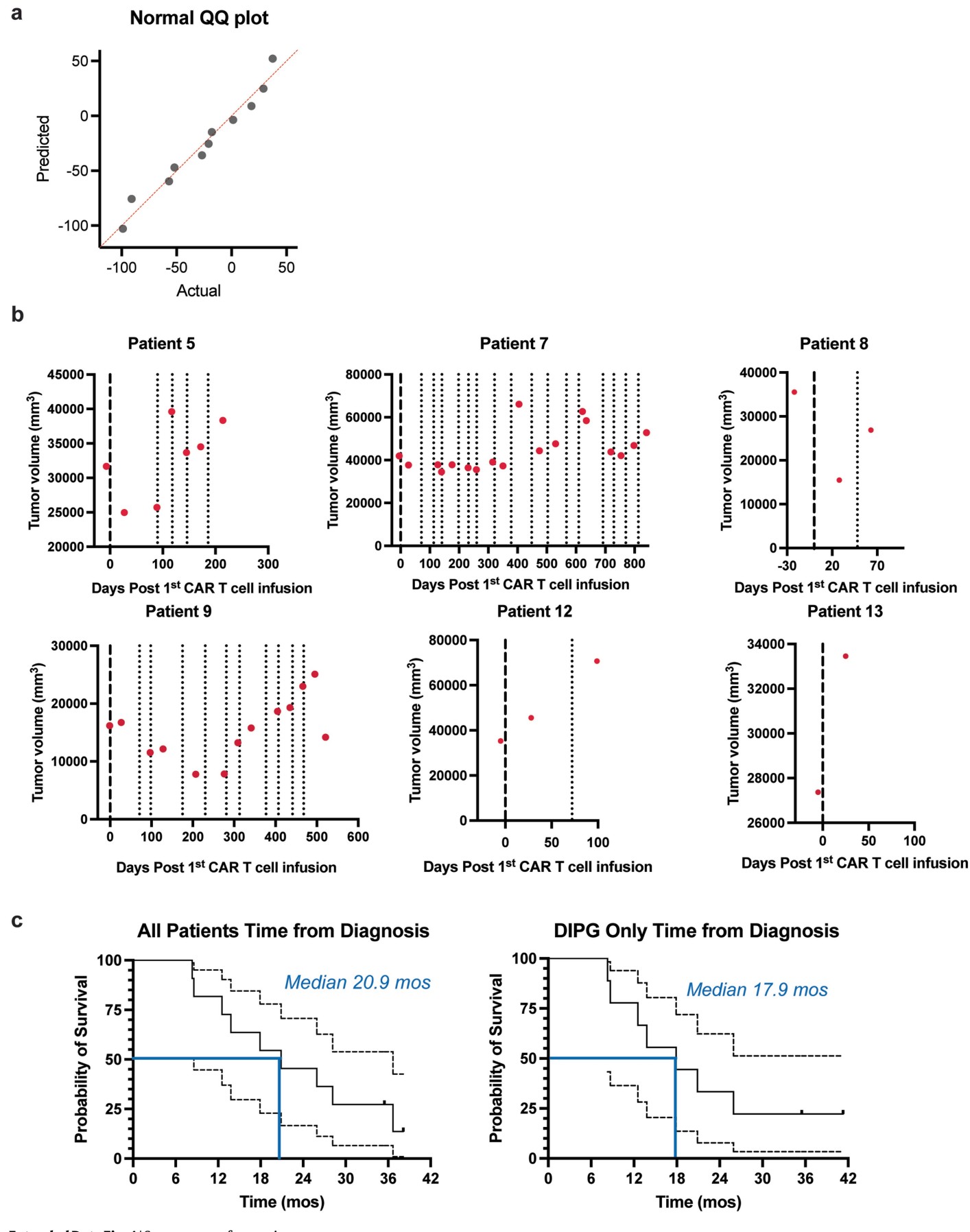

**Extended Data Fig. 4** | See next page for caption.

**Extended Data Fig. 4 | Tumor volumetric change over time and overall survival. a**. QQ plot of Shapiro-Wilk Test Normality Test for Gaussian Distribution (significance level alpha = 0.05). **b**. Tumor volume as a function of days following first GD2-CART infusion is shown using the same methods used in Figs. 2 and 3. Red dots represent timing of measurements; dashed line denotes IV GD2-CART infusion; dotted lines denote ICV GD2-CART infusions. Volumetric data for patients 1, 3 and 4 were presented previously in[2], denoted in that report as DIPG Patient #1, DIPG Patient #2 and DIPG Patient #3, respectively. Volumetric data for patients 6 and 10 are presented in Figs. 2 and 3. **c**. Kaplan-Meier survival curves depicting time from diagnosis to death or data cutoff for all patients (left graph) and those patients with DIPG (right graph, excluding the two patients with spinal cord DMGs). Brookmeyer-Crowley method was used to calculate confidence intervals of median survival time.

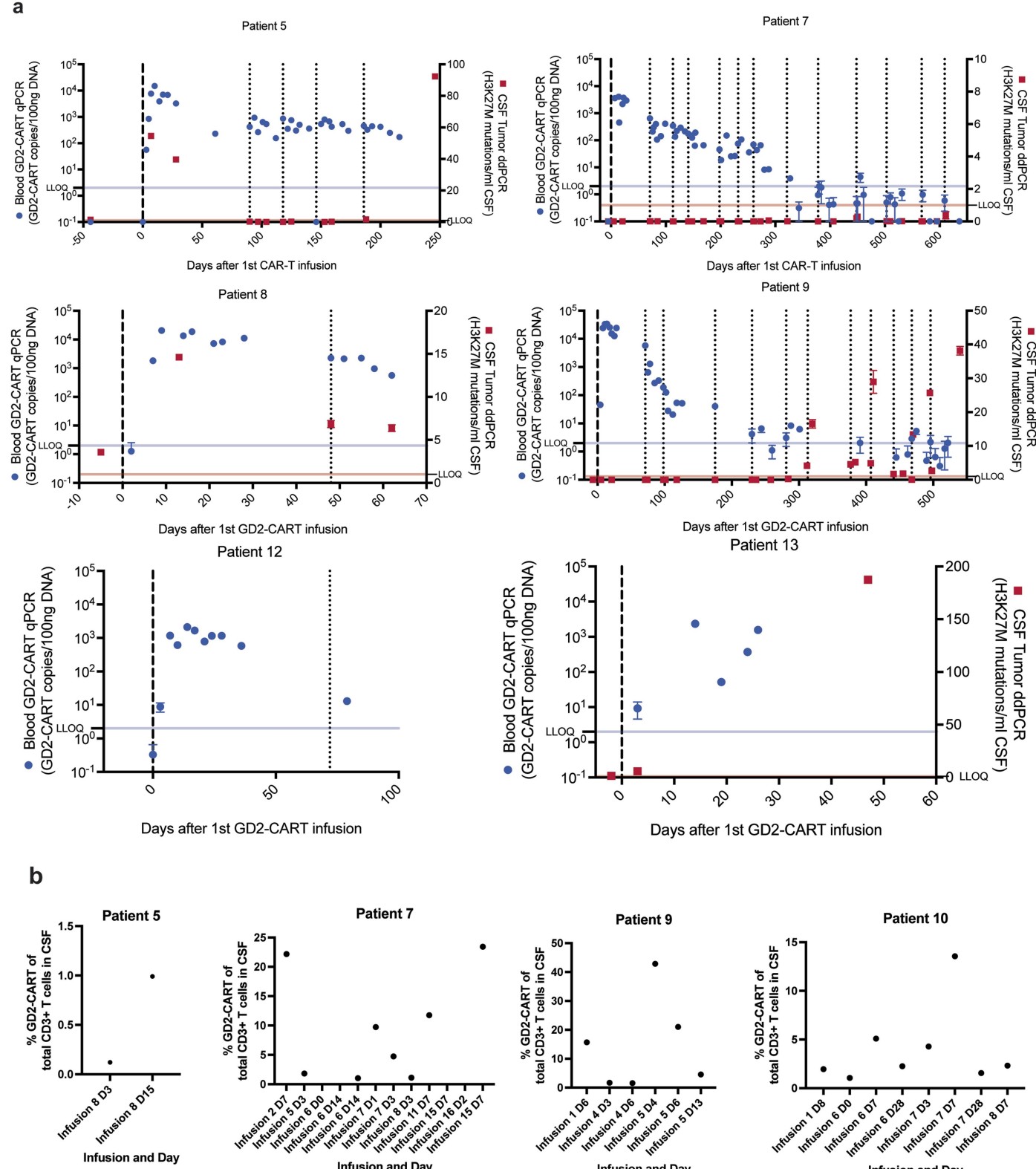

**a**

Patient 5 — Days after 1st CAR-T infusion

Patient 7 — Days after 1st CAR-T infusion

Patient 8 — Days after 1st GD2-CART infusion

Patient 9 — Days after 1st GD2-CART infusion

Patient 12 — Days after 1st GD2-CART infusion

Patient 13 — Days after 1st GD2-CART infusion

**b**

Patient 5 — Infusion and Day

Patient 7 — Infusion and Day

Patient 9 — Infusion and Day

Patient 10 — Infusion and Day

**Extended Data Fig. 5 | GD2-CART levels in blood and CSF, and tumor cell-free DNA in CSF. a**. GD2-CART copy number was detected using a validated qPCR method and based on a standard curve for GD2-CAR. The average GD2-CAR copy number were normalized to average albumin copies by plate. Copies per 100 ng DNA were depicted as mean ± standard deviation is depicted in blue. We quantified the H3K27M mutation copies per ml of cell free CSF using the ddPCR protocol established previously[2]; the mutation number/ml ± standard deviation is depicted in red. The dashed vertical lines designate infusion time points. The bold dashed vertical line designates the initial infusion. Each qPCR data point represents the mean of three technical replicates; each ddPCR data point represents the mean of four technical replicates; error bars, standard error of the mean. **b**. Percent of CD3 + T cells in CSF that are GD2-CAR-positive T cells. Flow cytometry (FACS) for GD2-CAR, using an anti-idiotype antibody (clone 1A7), demonstrated presence of GD2-CAR T cells (threshold of detection > 0.1% GD2-CART of CD3% T cells) in patient CSF following CAR T cell infusions in CSF samples with sufficient cells to assess by CAR-FACS.

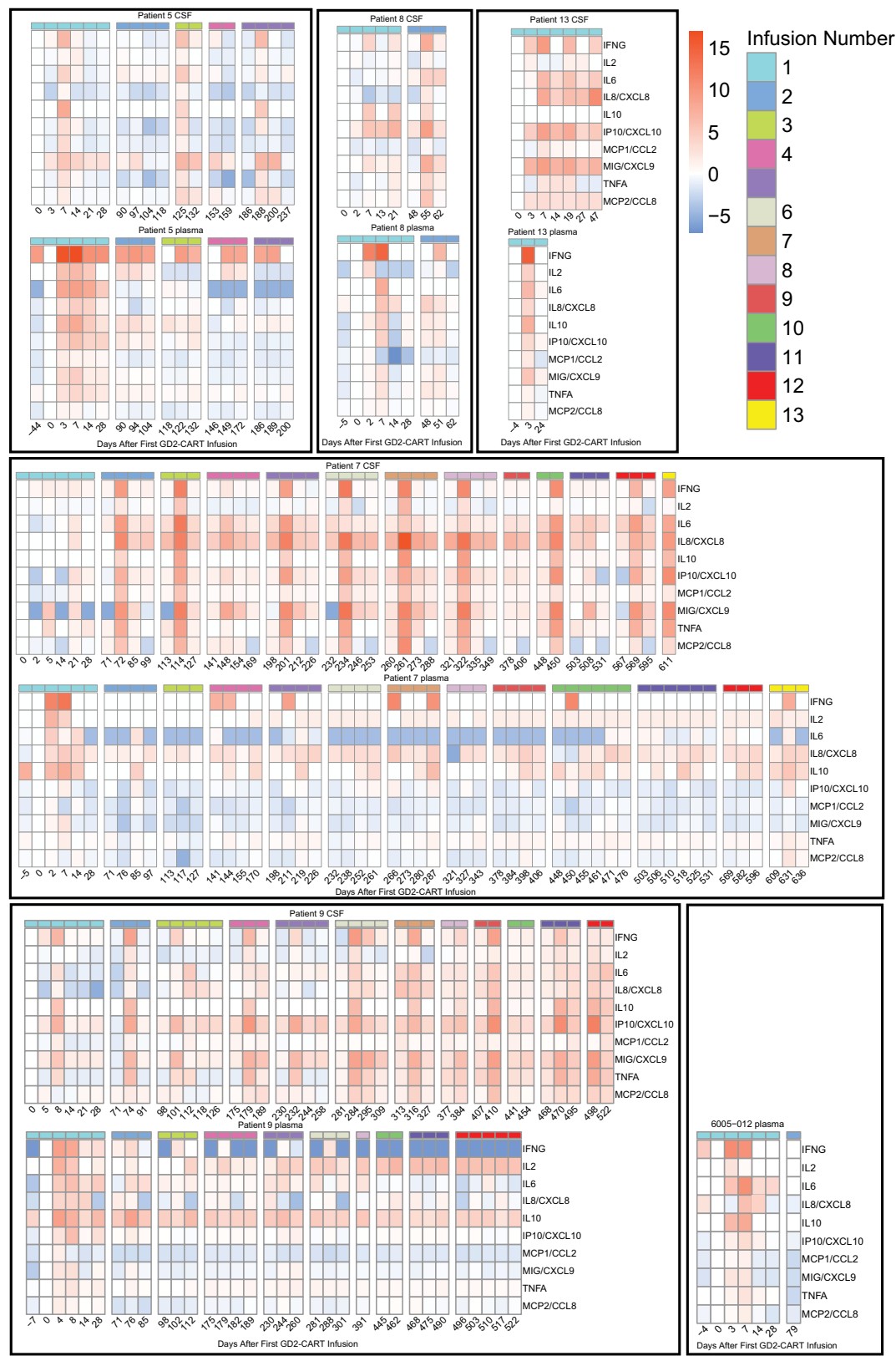

**Extended Data Fig. 6 | Plasma and CSF cytokine levels following GD2-CART infusions.** CSF cytokine levels after each infusion, expressed as log2 fold change from Day 0 prior to the first infusion for the indicated patient. Timepoints expressed as days from first infusion; (first infusion always administered i.v., subsequent infusions always administered i.c.v.); infusions are separated by a vertical white spacing to indicate the beginning of a new infusion cycle. Infusion number indicated by solid bar above each heatmap. Each square represents one measurement.

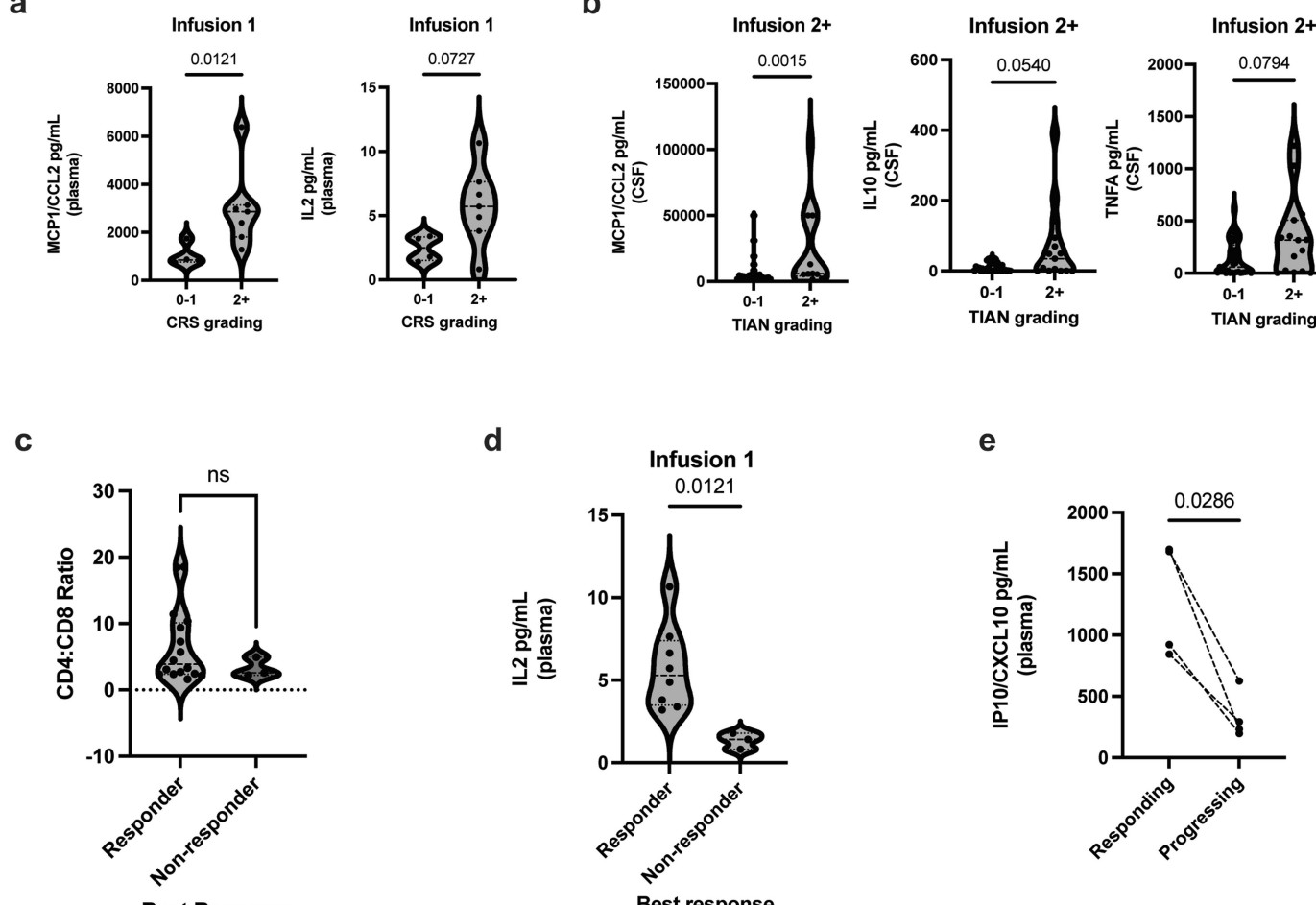

**Extended Data Fig. 7 | Correlates of toxicity and response. a**. Correlates of cytokine release syndrome (CRS). Higher plasma MCP-1 and IL2 levels correlate with grade 2 or higher CRS. (CRS grade 2+, n = 7 patients; CRS grade 0-1, n = 4 patients) **b**. Correlates of tumor inflammation-associated neurotoxicity (TIAN). Higher CSF levels of MCP-1, IL10 and TNF-alpha correlate with higher grade TIAN (grade 2 and higher). (TIAN grade 2+, n = 15 patients; TIAN grade 0-1, n = 33 patients) **c**. CD4:CD8 ratios were calculated based on manufacturing characterization and compared in patients based on best response (responder = decreased or stable tumor volume at best response and clinical improvement, n = 8 patients; non-responder = increased tumor volume at best response, n = 3 patients, as shown in Fig. 1b). **d**. Correlates of Response. Higher plasma levels of IL2 were found in patients who responded to GD2-CAR T cell therapy (n = 8 patients) compared to non-responders (n = 3 patients). **e**. Peak levels of IP10, a chemokine secreted in response to interferon-gamma

signaling, were decreased at the time of tumor progression compared to timepoints during response in the same patients; dotted line indicates paired values for individual patients (n = 3 patients with paired samples) at time of response (first ICV infusion) and time of progression. Cytokine/chemokine data is represented in pg ml−1 based on standard curves. For statistical analysis, data were grouped by patient, infusion, sample type, and cytokine. The maximum cytokine level per infusion for each patient was compared across a. CRS grading (Grade 0-1 vs Grade 2+), b. TIAN grading (Grade 0-1 vs Grade 2+), c-d. best response (responder = decreased or stable tumor volume at best response; non-responder = increased tumor volume at best response), and e. progression (responding = first ICV infusion; progressing = first infusion after which tumor progression was evident) using a two-tailed Mann-Whitney U Test. As these analyses were exploratory and sample size was limited, no multiple comparison analysis was conducted.

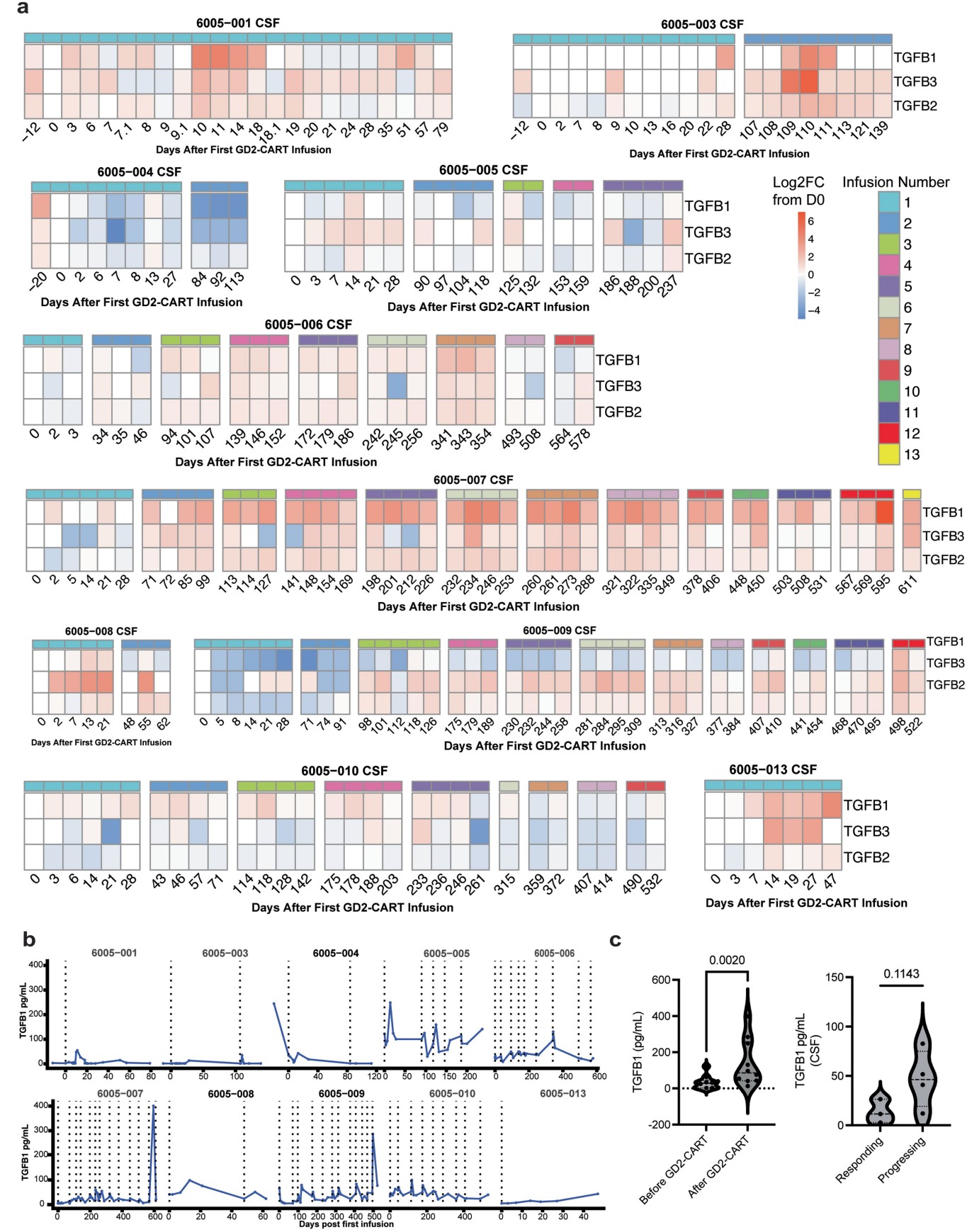

**Extended Data Fig. 8** | See next page for caption.

**Extended Data Fig. 8 | CSF TGF-beta cytokine levels following GD2-CAR T infusions. a**. CSF TGF-beta cytokine levels after each infusion, expressed as log2 fold change from Day 0 prior to the first infusion for the indicated patient. Timepoints expressed as days from first infusion; (first infusion always administered i.v., subsequent infusions always administered i.c.v.); infusions are separated by a vertical white spacing to indicate the beginning of a new infusion cycle. Infusion number indicated by solid bar above each heatmap. Each square represents one measurement. **b**. Absolute TGF-beta1 levels per patient over time. Dotted line represents each GD2-CAR T infusion for each patient. Data for Patient 012 is missing due to limited CSF sampling for that individual. **c**. TGF-beta1 levels before and after GD2-CAR T cell therapy (left, n = 10 patients) and at early timepoints following GD2-CAR T infusion before ("responding", n = 3 patients) and after progression (right, n = 4 patients). Statistical comparison by two-tailed Mann-Whitney U Test.

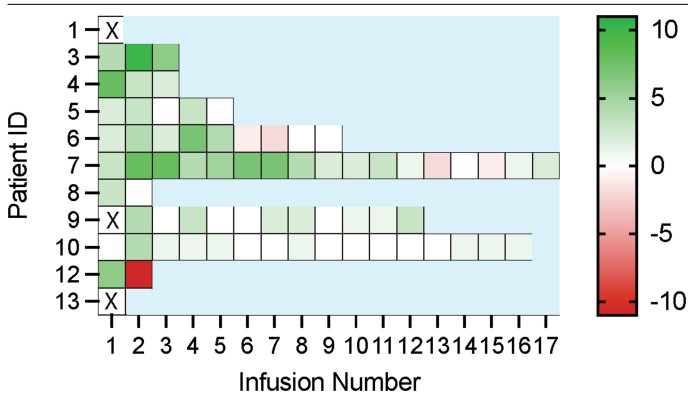

**Extended Data Fig. 9 | Clinical Improvement Scores.** The clinical improvement scale (CIS) is simple tool to quantify changes in the neurological exam[2] in which each item tested in a comprehensive neurological exam is assigned one point if improved, and −1 point if worse than pre-infusion baseline for each infusion. Clinical improvement scores are assessed one month after each infusion, and compared to the pre-infusion baseline for that infusion. The clinical improvement scale was not assessed if the patient required corticosteroid therapy until at least 7 days from cessation of corticosteroids to avoid confounding effects on neurological function. Each GD2-CAR T cell infusion is treated independently, such that the changes reported compared to the pre-infusion baseline for that infusion. A score of zero (represented as white here) means that there was no clinical change, or that there were an equal number of improved and worsened symptoms/signs following that infusion. Please note that, because the score reflects change after a given infusion, neurological improvement after a previous cycle, with continued benefit but no further improvement, would be annotated with white (no change) in this heatmap. Similarly, improvement in one domain with decreased function in another domain, would also be annotated with white (no change) in this heatmap. Green = better than pre-infusion baseline for that infusion, red = worse than pre-infusion baseline for that infusion. An X indicates that the CIS was not evaluable due to corticosteroid use.

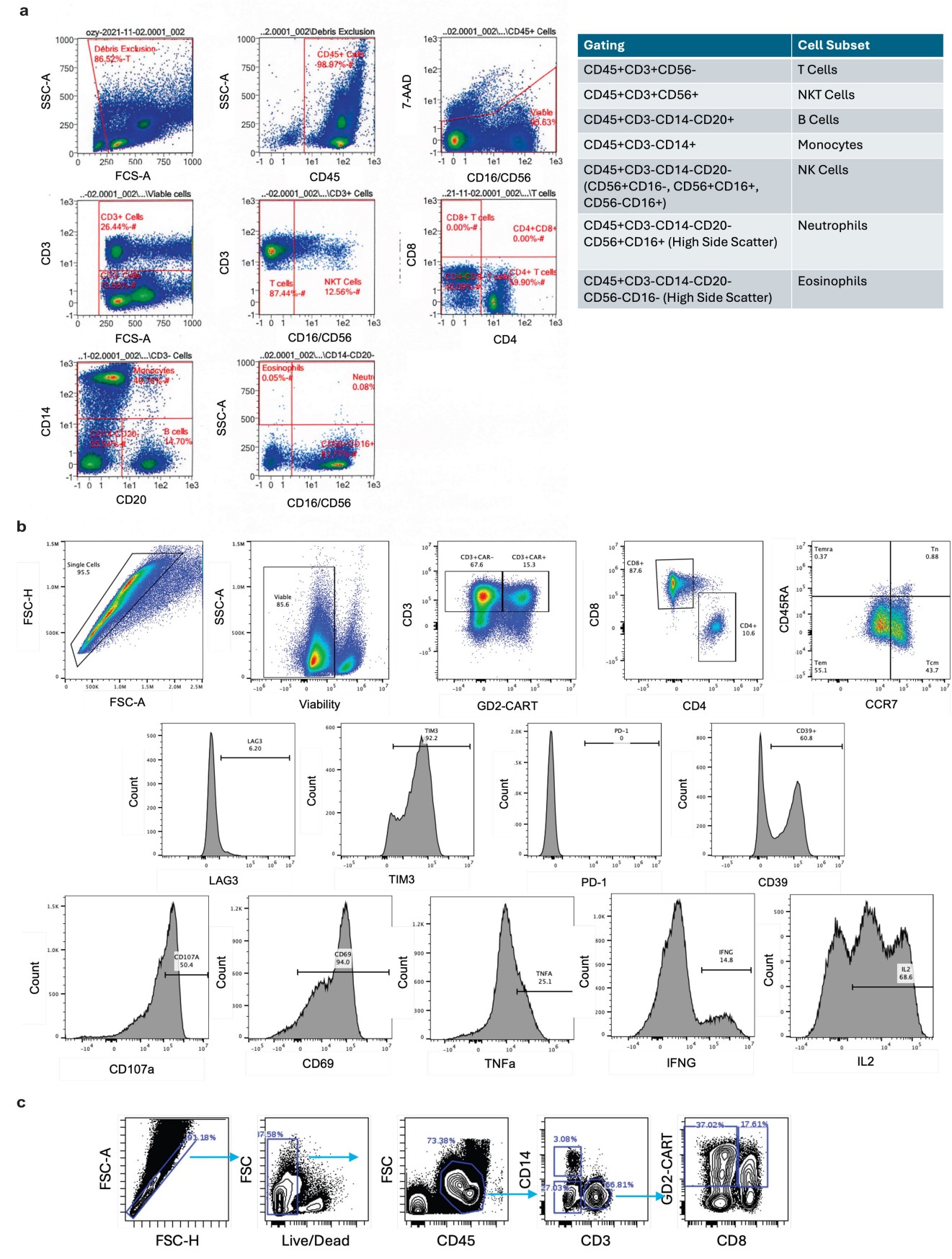

**Extended Data Fig. 10 | Representative gating for flow cytometry of patient samples. a**. Representative gating of immune subsets throughout manufacturing process. **b**. Representative gating of phenotypic and functional analyses of manufactured GD2-CART following manufacturing. **c**. Representative gating of a patient CSF sample by flow cytometry (identical gating was used for PBMC obtained from blood).

| --- | --- |

# Reporting Summary

## Statistics

For all statistical analyses, confirm that the following items are present in the figure legend, table legend, main text, or Methods section.

| n/a | Confirmed | |
| --- | --- | --- |
| ☐ | ☒ | The exact sample size (*n*) for each experimental group/condition, given as a discrete number and unit of measurement |
| ☐ | ☒ | A statement on whether measurements were taken from distinct samples or whether the same sample was measured repeatedly |
| ☐ | ☒ | The statistical test(s) used AND whether they are one- or two-sided<br>*Only common tests should be described solely by name; describe more complex techniques in the Methods section.* |
| ☐ | ☒ | A description of all covariates tested |
| ☐ | ☒ | A description of any assumptions or corrections, such as tests of normality and adjustment for multiple comparisons |
| ☐ | ☒ | A full description of the statistical parameters including central tendency (e.g. means) or other basic estimates (e.g. regression coefficient) AND variation (e.g. standard deviation) or associated estimates of uncertainty (e.g. confidence intervals) |
| ☐ | ☒ | For null hypothesis testing, the test statistic (e.g. *F*, *t*, *r*) with confidence intervals, effect sizes, degrees of freedom and *P* value noted<br>*Give P values as exact values whenever suitable.* |
| ☒ | ☐ | For Bayesian analysis, information on the choice of priors and Markov chain Monte Carlo settings |
| ☒ | ☐ | For hierarchical and complex designs, identification of the appropriate level for tests and full reporting of outcomes |
| ☒ | ☐ | Estimates of effect sizes (e.g. Cohen's *d*, Pearson's *r*), indicating how they were calculated |

*Our web collection on statistics for biologists contains articles on many of the points above.*

## Software and code

Policy information about availability of computer code

| Data collection | n/a |
| --- | --- |
| Data analysis | PRISM (version 10) software was used for statistical analyses |

For manuscripts utilizing custom algorithms or software that are central to the research but not yet described in published literature, software must be made available to editors and reviewers. We strongly encourage code deposition in a community repository (e.g. GitHub). See the Nature Portfolio guidelines for submitting code & software for further information.

## Data

Policy information about availability of data

All manuscripts must include a data availability statement. This statement should provide the following information, where applicable:

- Accession codes, unique identifiers, or web links for publicly available datasets
- A description of any restrictions on data availability
- For clinical datasets or third party data, please ensure that the statement adheres to our policy

| All raw data are available in the source data file. |
| --- |

# Research involving human participants, their data, or biological material

Policy information about studies with human participants or human data. See also policy information about sex, gender (identity/presentation), and sexual orientation and race, ethnicity and racism.

| | |
|---|---|
| Reporting on sex and gender | Of the 13 patients enrolled, 7 patients were female and 6 patients were male |
| Reporting on race, ethnicity, or other socially relevant groupings | n/a |
| Population characteristics | Enrollment began in June 2020 and the data cutoff was December 1, 2023.  Median age was 15 yrs (range 4-30 yrs), and seven patients were female. Ten patients had DIPG and three had sDMG. Two patients (Patients 002 and 011) were removed from study prior to treatment due to rapid tumor progression and a decline in performance status rendering them ineligible to protocol-directed therapy. |
| Recruitment | Patients on this Phase 1 clinical trial were recruited through physician and self-referral. PAtents were recruited from throughout the United States, and referrals came from both academic centers and community health centers. |
| Ethics oversight | The Stanford University IRB approved this clinical study. |

Note that full information on the approval of the study protocol must also be provided in the manuscript.

# Field-specific reporting

Please select the one below that is the best fit for your research. If you are not sure, read the appropriate sections before making your selection.

☒ Life sciences  ☐ Behavioural & social sciences  ☐ Ecological, evolutionary & environmental sciences

For a reference copy of the document with all sections, see nature.com/documents/nr-reporting-summary-flat.pdf

# Life sciences study design

All studies must disclose on these points even when the disclosure is negative.

| | |
|---|---|
| Sample size | Arm A followed a 3+3 dose escalation design, starting at IV Dose Level 1 (IV DL1). Disease cohorts were analyzed separately, but the safety in subjects with DIPG informed dose escalation for subjects with spinal DMG, but not vice-versa. The rationale for this was concerns about enrollment speed (pediatric spinal DMG is less common than pediatric DIPG) and lower risk for CAR-inflammation associated brain herniation (due to the location of their tumor). The study also allowed up to 6 subjects to be replaced as inevaluable because they were unable to meet the protocol-defined eligibility to receive GD2CART cells to account for significant disease progression post-enrollment and prior to treatment. In total, Arm A enrolled 11 subjects (with 11 successful manufacturing runs) and treated 7. Four subjects (3 subjects with DIPG, 1 subject with spinal DMG) were enrolled on IV DL1 and three subjects treated, with 0 of 3 subjects with DIPG experiencing DLT. Because safety in DIPG informed spinal DMG cohort, both groups dose escalated to IV DL2. In IV DL2, 1 of 3 subjects with DIPG experienced DLT, necessitating expansion to 6 subjects with DIPG. Concurrent enrollment of subjects with spinal DMG to IV DL2 resulted in 1 of 2 subjects with spinal DMG experiencing DLT. Ultimately, 2 of 6 subjects with DIPG and 1 of 2 subjects with spinal DMG experienced DLT, causing IV DL2 to exceed allowable DLT rate.  All DLT cases were due to cytokine release syndrome (CRS).  All inevaluable subjects in Arm A were ineligible to receive GD2CART cells on study because of disease progression. Because CRS caused by systemic immune activation was cause of all DLT, we subsequently modified the protocol and began to test an ICV-only strategy route and schedule (Arms B and C) that would diminish the chances of dose-limiting CRS. |
| Data exclusions | No data were excluded for the 11 patients treated on trial. Two patients (Patients 002 and 011) were removed from study prior to treatment due to rapid tumor progression and a decline in performance status rendering them ineligible to protocol-directed therapy. One of these two patients was treated on an eIND and reported in Majzner et al., 2022 Nature. |
| Replication | ctDNA, and RT-PCR for CAR transgene were performed in triplicate, Cytokine analyses were performed in duplicate. |
| Randomization | No randomization was performed in this Phase 1 clinical trial, as this is a non-randomized early-phase trial. |
| Blinding | No blinding was performed in this Phase 1 clinical trial, as this is a non-blinded early-phase  trial. |

# Reporting for specific materials, systems and methods

We require information from authors about some types of materials, experimental systems and methods used in many studies. Here, indicate whether each material, system or method listed is relevant to your study. If you are not sure if a list item applies to your research, read the appropriate section before selecting a response.

## Materials & experimental systems

| n/a | Involved in the study |
|---|---|
| ☐ | ☒ Antibodies |
| ☒ | ☐ Eukaryotic cell lines |
| ☒ | ☐ Palaeontology and archaeology |
| ☒ | ☐ Animals and other organisms |
| ☐ | ☒ Clinical data |
| ☒ | ☐ Dual use research of concern |
| ☒ | ☐ Plants |

## Methods

| n/a | Involved in the study |
|---|---|
| ☒ | ☐ ChIP-seq |
| ☐ | ☒ Flow cytometry |
| ☐ | ☒ MRI-based neuroimaging |

## Antibodies

| Antibodies used | Each row provides information in the following order: Antigen Fluorochrome Clone Supplier Part Number<br>CD3 FITC UCHT1 BioLegend 300406<br>CD8 PerCP Cy5.5 SK1 BD Pharmingen 565310<br>CD45 BV785 2D1 BioLegend 368528<br>CD4 BV711 RPA-T4 BioLegend 300558<br>CD95 BV650 DX2 BioLegend 305624<br>CD39 Bv605 A1 BioLegend 328236<br>Cell viability BV510 N/A Invitrogen L-34965<br>CD57 BV421 NK-1 BDBiosciences 563896<br>CCR7 BUV805 2L1A BDBiosciences 749673<br>CD45RA Alx700 HI100 BioLegend 304120<br>GD2CAR DyLight650 1A7  NCI Biological Resources Branch, Custom<br>CD14 PE-Cy7 63D3 BioLegend 367112<br>CD11b APC-Cy7 ICRF44 BioLegend 301352<br>CD33 PE-Dazzle WM53 BIolegend 303432<br>GD2 PE 14G2A BioLegend 357304 |
|---|---|
| Validation | All antibodies were validated as reported by the manufacturer (references below) except 1A7. In the case of 1A7, the anti-GD2 CAR idiotype antibody, untransduced T cells were used as a biological control. To Determine the optimal concentration of antibody for staining in the CAR-FACS panel, each fluorochrome-conjugated antibody was titrated to determine the saturating amount of antibody needed to stain a test/ one million cells. Dose response curves for each antibody informed the saturating amount of fluorochrome-conjugated antibody for staining a million cells in 100 ml staining volume.<br>antigen and antibody validation reference:<br>CD3 Salmeron A et al.1991<br>CD8 Behjat et al. 2005<br>CD45 Knapp W et al.1989<br>CD4 Knapp W et al.1989<br>CD95 Schlossman S 1995<br>CD39 Aversa G. 1988<br>CD57 Abo T.  et al.1981<br>CCR7 Birkenbach M. 1993<br>CD45RA Knapp W et al.1989<br>GD2-CAR Sen et al. 1997<br>CD14 Antonyshyn et al. 2022<br>CD11b Knapp W et al.1989<br>CD33 Knapp W et al.1989<br>GD2 Mujoo et al. 1989 |

## Clinical data

Policy information about clinical studies

All manuscripts should comply with the ICMJE guidelines for publication of clinical research and a completed CONSORT checklist must be included with all submissions.

| Clinical trial registration | NCT04196413 |
|---|---|
| Study protocol | The trial protocol is provided in the supplementary information |
| Data collection | June 2020 to December 2023 at the Lucile Packard Children's Hospital at Stanford. |
| Outcomes | Primary objectives were to determine feasibility of manufacturing, assess safety and tolerability and identify a maximally tolerated and/or recommended phase 2 dose of IV GD2-CART following lymphodepleting chemotherapy in this population. Secondary objectives included preliminary assessment of benefit as measured by radiographic response on MRI and by clinical improvement. Another secondary objective was to assess the safety and  benefit of subsequent ICV administrations of GD2-CAR T-cells |

## Plants

| | |
|---|---|
| Seed stocks | n/a |
| Novel plant genotypes | n/a |
| Authentication | n/a |

## Flow Cytometry

### Plots

Confirm that:

☒ The axis labels state the marker and fluorochrome used (e.g. CD4-FITC).

☒ The axis scales are clearly visible. Include numbers along axes only for bottom left plot of group (a 'group' is an analysis of identical markers).

☒ All plots are contour plots with outliers or pseudocolor plots.

☒ A numerical value for number of cells or percentage (with statistics) is provided.

### Methodology

| | |
|---|---|
| Sample preparation | PBMC were isolated from fresh whole blood by gradient centrifugation on ficoll (Ficoll paque Plus, GE Healthcare, SigmaAldrich). Two to five million PBMC were stained with fixable Live/Dead aqua (Invitrogen) amine-reactive viability stain. Cells were then preincubated with Fc block (trustain, Biolegend) for 5 min, then stained at room temperature with the following fluorochrome conjugated mAb in an 15-color, 17-parameter staining combination. |
| Instrument | LSR (BD Biosciences) |
| Software | Analysis was performed in FlowJo |
| Cell population abundance | At least 106 cells were acquired unless restricted by the number of cells isolated from 8 ml of whole blood or when acquiring CSF isolated cells. The assay limit of detection for cells calculated as 1 in 104 of total acquired PBMCs. |
| Gating strategy | CAR T-cells in CSF and PBMC: Singlets->viable cells->CD45+->CD14-, CD3+ -> CD4 or CD8<br>CAR positivity gated based on control PBMC.<br>Representative gating strategy is shown in Extended Data Figure 10. |

☒ Tick this box to confirm that a figure exemplifying the gating strategy is provided in the Supplementary Information.

## Magnetic resonance imaging

### Experimental design

| | |
|---|---|
| Design type | clinical studies |
| Design specifications | clinical studies |
| Behavioral performance measures | n/a |

### Acquisition

| | |
|---|---|
| Imaging type(s) | clinical studies |
| Field strength | 3T |
| Sequence & imaging parameters | T2 sequences used for volumetric quantification of tumor size and T2 sequences shown |
| Area of acquisition | Brain and Spine |
| Diffusion MRI | ☐ Used  ☒ Not used |

## Preprocessing

| | |
|---|---|
| Preprocessing software | n/a |
| Normalization | n/a |
| Normalization template | n/a |
| Noise and artifact removal | n/a |
| Volume censoring | *Define your software and/or method and criteria for volume censoring, and state the extent of such censoring.* |

## Statistical modeling & inference

| | |
|---|---|
| Model type and settings | n/a |
| Effect(s) tested | n/a |

Specify type of analysis: ☒ Whole brain ☐ ROI-based ☐ Both

| | |
|---|---|
| Statistic type for inference<br>(See Eklund et al. 2016) | n/a |
| Correction | n/a |

## Models & analysis

| n/a | Involved in the study |
|---|---|
| ☒ ☐ | Functional and/or effective connectivity |
| ☒ ☐ | Graph analysis |
| ☒ ☐ | Multivariate modeling or predictive analysis |

