## [Peer Review File · Nature]

Manuscript Title: IV and intracranial GD2-CAR T-cells for H3K27M+ diffuse midline gliomas

Reviewer Comments & Author Rebuttals

Reviewer Reports on the Initial Version:

Referees' comments:

Referee #1 (Remarks to the Author):

The manuscript entitled “Sequential intravenous and intracerebroventricular GD2-CAR T-cell therapy for H3K27Mmutated diffuse midline gliomas,” reports findings from Arm A of a multi-arm clinical trial evaluating GD2-CAR T cell therapy for the treatment of DMG. This report summarizes clinical trial results for manufacturing feasibility, therapeutic tolerability, and clinical activity in 11 patients treated with intravenous followed by intracerebroventricular GD2-CAR T cell therapy, extending this team’s prior report previously published in Nature (2022), which detailed four patients treated on DL1 (three of whom are included in this manuscript).

This is a well-presented and important clinical study showing early promising findings for the therapeutic benefit of GD2-CAR T cell therapy against DIPG/DMG, with clinical response/neurological benefit noted in 8 of 11 treated patients, including an ongoing complete response at 30 months. This study also defines an MTD for i.v. administered GD2-CAR T cell therapy and provides an initial comparison of i.v. versus i.c.v. administered CAR T cells. The correlative assessments presented were somewhat limited and expanding the analysis of the correlative findings in this cohort would have strengthened the manuscript.

Comments:

1. It would be helpful if Authors could more clearly identify which of the patients were previously included in the Nature 2022 report, since the numbering is different. While some of this information is provided in the Figure 1 legend and Table 1, the numbering for the prior DIPG Patient 1 should also be specifically noted. The consort diagram lists the “compassionate treatment off-study.” If this is the prior DMG patient 1/002 who is not presented in the manuscript, please also specifically note this in the manuscript.
2. Figure 1 legend states that two patients (Patients 006 and 009) received re-irradiation however the swimmer’s plot shows that 007 also was re-irradiated. Please address this discrepancy. Please also comment in the Results on the re-irradiation and if any other treatments were allowed while on-study.
3. The criteria for being on-study should be better described in the manuscript, not just in the clinical protocol provided in the supplement. It was unclear from Fig 1c if the blue (on-study time) represents

the length of time of clinical response/improvement for each patient. Please clarify this point, and if it doesn't, it would be important to add this for all patients.

4. Were toxicities – either CRS, ICANS or TIAN -- associated with tumor burden at time of treatment or tumor location or i.c.v. infusion dose. Please comment in the manuscript.

5. The volumetric change in tumor size for many of the patients, particularly the spinal DMG, are impressive. Please note what timepoint the 'best radiographic volume change" was achieved, possibly by indicating on the patient-specific tumor volume plots

6. For Figure 2d (and Fig 3d and supplemental figs) it is unclear how many days after the i.c.v. infusion cytokines were measured. Is the first time point (day 43, 114, 176 etc) pre or post i.c.v. administration?

7. While the authors' restraint in concluding a survival benefit is appreciated, median survival times and confidence intervals for the Kaplan Meier plots (Supp Fig 4) should be reported. It would also seem relevant to report overall survival from time of treatment.

8. Are there any learnings from the cytokine changes in the plasma or CSF that can be gleaned regarding toxicity, response, or progression?

Referee #2 (Remarks to the Author):

Patients with DIPG/DMG have a median overall survival of 11 months – It is a cancer of the most dismal prognosis and new treatments are urgently required. H3K27M-mutant diffuse midline gliomas (DMGs) express high levels GD2 at the cell surface, making GD2-CAR T-cells, a promising therapy. Authors have previously reported the first three patients treated at dose level #1 and this manuscript is the final clinical results.

In this second and final report from the clinical trial (NCT04196413), the authors present compelling evidence for using GD2 CAR T cell immunotherapy for DMG.

An intravenous CART cell dose was administered after lymphodepleting chemotherapy. Patients who responded positively were offered subsequent intracerebroventricular (ICV) infusions.

Thirteen patients joined the trial, and eleven received the treatment. The cells were successfully manufactured for all patients, in an average of 7 days. At the lower dose level, there were no serious side effects, but at the higher dose, three patients had a severe immune reaction called cytokine release syndrome. Nine patients also received additional doses directly into their brain fluid, which did not cause severe side effects, but $1e6/kg$ was established as the maximally tolerated dose IV.

All patients experienced some level of brain inflammation, but four patients had significant reductions in

their tumour size, and astonishingly one patient's tumour completely disappeared for over 30 months. Figure 2 shows patient 10's complete response, with a disappearing tumour volume (Fig2b) and an absence of detecting H3K27M mutations in CSF via ddPCR (Fig2c). The response is quite extraordinary. For this devastating tumour type, this is an incredibly exciting result. Eight patients showed improvement in their neurological symptoms.

There was excellent supportive care, toxicity monitoring, clinical management and response assessment in place. The brain inflammation side effects were managed carefully with close monitoring and a meticulous treatment plan.

In general clinical improvement is tracked with tumour response on MRI. ICANS was rarely seen and observed only after IV administration but not with ICV infusions, which provides evidence that intracranial delivery of CAR T cells to treat brain tumours is safe. The TIAN seen was transient and manageable. Interestingly, the CAR construct contained an inducible caspase-9, but the agent available to ablate the CAR T cells via Caspase-9 induction (AP1903) was not administered to any patients treated on Arm A.

The well-written manuscript has no major flaws prohibiting its publication and it is a novel study. The statistical tests applied are sound. Overall, the treatment shows immense promise in reducing tumours and improving symptoms in these patients, a cohort who currently have no other options. Future trials will move to intracranial ICV delivery only and explore the role of lymphodepleting chemotherapy. The demonstration of such positive CART cell responses against such an insidious solid tumour in the CNS is exciting. The report is likely to be highly cited in the field.

Major Points:

- Given the differential responses in patients and the diversity in product delivered, it would be beneficial for authors to include, if possible, the stratification of the proportion of CD4 or CD8 cells, of the CD3+ CART+ population that persisted in the patients. Did patients #6 and #10 have a differential skewing of CD4/CD8 ratio?
- T cells can downregulate cell surface CAR after activation. The authors have used two methods to determine CAR T cell persistence and report PCR as a read-out for CAR T cell persistence in the main figures 2C and 3C. However, the authors measured cell surface CAR expression (as shown in Sup Fig 3e, presumably using an anti-idiotypic antibody, as is described in Supp Fig 5b) – with varying results. Can the authors reconcile why the cell surface expression seems variable, and they have elected to display PCR as a readout of CAR expression? Note also the anti-CAR staining is missing from the supplementary methods.

Minor Points:

- Supp Fig 1e. It is unclear in the figure legend what the n numbers are representing on the graph – There are 19 data points but only 11 patients were reported in this study – Where do the other 8 data points come from? Similarly, it is unclear what the data points represent in Supp Fig1D (assuming a minority of patients had CD4/CD8 ratios determined by flow at apheresis but n should be stipulated in legend for all panels. I assume Supp Fig1F is the characterisation of drug product on the day of

administration but this too should be clarified in legend.

- There are some typesetting issues in the clinical trial protocol in the supplementary figures where some embedded links have been formatted incorrectly in the final PDF (Example page 41/161 and bottom of page 58) and some small typos (Page 43/161: “... Were consistently filters when put through ...” Which are tiny but authors may want to correct.
- A couple of typos in the manuscript (Page 10, pseudoprogession)

Referee #3 (Remarks to the Author):

Monje et al. present a Phase 1 study of T cells expressing a 2nd generation GD2-specific CAR (CD8TM,41BB costim) administered first IV followed by ICV injection on two dose levels in patients with H3K27M DIPGs and sDMGs. Cells were manufactured by the Miltenyi Prodigy system in 7 days with IL7/15 and were enriched for central memory subset. The manuscript outlines safety parameters and defines the MTD. Objective responses are demonstrated by RANO2 criteria in 4/11 patients and additional four patients had meaningful clinical benefit as measured by the team’s standardized clinical improvement score. The team provides data on CAR T cell kinetics in peripheral blood and ctDNA in CSF, latter nicely demonstrating responses in addition to imaging and clinical findings. Interestingly DLTs occurred in the context of IV CAR T cell administration, but were manageable for ICV administration; thus, prompting the team to switch to ICV injections for future studies.

The manuscript is a major milestone in our understanding of CAR T cells’ safety and efficacy to treat patients with DIPG/sDMG. The manuscript is overall well written, but could benefit from additional analyses as outlined below:

1. The incidence and severity of CRS at $3 \times 10^6/\text{kg}$ IV dose is surprising. Table 1 summarizes these toxicities, but would benefit from more in-depth assessment and reporting.

- Can the authors provide more detail on what contributed to CRS on the IV dose? Specifically, the components of the CRS grading system would be informative to report: fever, degree of O2 requirement, pressor support. How do these IV CAR T cell mediated toxicities compare to Buffalo et al NEJM 2023 or Lin et al JCO 2024? Did the team notice peripheral nerve related toxicities after IV CAR T cell administration?

- Fig 2 and 3 shows data from two selected patients for cytokines, but it would important to gain a general view from the treated patients and compare the findings to previously reported cytokine profiles. Supplementary table 2 outlines TIAN mgmt but is missing information on CRS mgmt; the inducible caspase 9 was fortunately not needed. What mgmt strategies were employed for CRS (some may overlap with TIAN/ICANs)?

- Are there differences in specific CSF cytokine/chemokine levels in responders vs non-responders?

2. On the consort diagram, 83 patients were evaluated, 34 were excluded due to not meeting eligibility criteria and 13 were enrolled. Please, clarify what happened to patients meeting eligibility criteria, but not enrolled.

3. CAR T cell expansion and persistence related assessments:

- Some of the peripheral blood kinetics are shown in Fig 2 & 3 (and in suppl figs), but these are two selected patients and it is important to provide an overview of this critical parameter for all patients as a main figure. A similar graph with ctDNA would be highly informative especially if expansion/persistence can be correlated with drop in ctDNA.

- What is the expansion and persistence of CAR T cells in CSF? This parameter seems to be missing from the manuscript.

4. One of the key findings from Majzner et al was the demonstration of antitumor activity of ICV dosing. With the number of patients treated now, can the authors summarize and comment on benefit of repeat infusions after IV dosing (beyond increased safety).

Minor:

Could you clarify the number of patients with clinical and imaging responses. The abstract lists four + four, but the text states “nine experienced clinical or imaging benefit”.

The product is enriched for CD4 cells although the cytokines used should promote CD8s. Is CD4 promoting effect related to desatinib?

Suppl. Fig 1e and 3: there are more measurements on the graphs than the number of patients. Are some of these duplicates? Please, clarify.

The manuscript text states max infusion of 16, but the figure 1 seems to show 17. Please, clarify.

What do the asterisks represent in Fig 1c?

Author Rebuttals to Initial Comments:

We were delighted to see the enthusiastic comments and thorough, thoughtful suggestions. We have now addressed each comment with clarifying edits to the manuscript and/or figures, and have added new analyses of cytokine/chemokine correlates of toxicity and response, as detailed in the point-by-point response below. These changes have improved the manuscript, and we are grateful for the constructive input.

Point-by-point response to referees' comments

Referee #1

The manuscript entitled “Sequential intravenous and intracerebroventricular GD2-CAR T-cell therapy for H3K27Mmutated diffuse midline gliomas,” reports findings from Arm A of a multi-arm clinical trial evaluating GD2-CAR T cell therapy for the treatment of DMG. This report summarizes clinical trial results for manufacturing feasibility, therapeutic tolerability, and clinical activity in 11 patients treated with intravenous followed by intracerebroventricular GD2-CAR T cell therapy, extending this team’s prior report previously published in Nature (2022), which detailed four patients treated on DL1 (three of whom are included in this manuscript).

This is a well-presented and important clinical study showing early promising findings for the therapeutic benefit of GD2-CAR T cell therapy against DIPG/DMG, with clinical response/neurological benefit noted in 8 of 11 treated patients, including an ongoing complete response at 30 months. This study also defines an MTD for i.v. administered GD2-CAR T cell therapy and provides an initial comparison of i.v. versus i.c.v. administered CAR T cells. The correlative assessments presented were somewhat limited and expanding the analysis of the correlative findings in this cohort would have strengthened the manuscript.

We thank the referee for the positive and supportive comments.

We appreciate the comment about the value of further correlative studies, and have now included more correlative findings describing associations of cytokine/chemokines with toxicity and response in this patient cohort.

We have also performed extensive single cell sequencing studies that we are in the midst of analyzing and are also in the midst of conducting preclinical experimental validation of hypotheses generated by the single cell analyses. This is a work-in-progress, and this enormous amount of clinical correlative and preclinical data (including > 1 million single cells from patient samples) will require a dedicated follow-on paper to describe.

Comments:

1. It would be helpful if Authors could more clearly identify which of the patients were previously included in the Nature 2022 report, since the numbering is different. While some of this information is provided in the Figure 1 legend and Table 1, the numbering for the prior DIPG Patient 1 should also be specifically noted. The consort diagram lists the “compassionate treatment off-study.” If this is the prior DMG patient 1/002 who is not presented in the manuscript, please also specifically note this in the manuscript.

This is an excellent point; we have now updated Table 1 to more clearly indicate which patients were previously reported in Majzner et al (including Patient 001/”DIPG Pt 1”). A sentence has been added to the main manuscript text describing treatment of Patient 002 on a single patient compassionate IND.

2. Figure 1 legend states that two patients (Patients 006 and 009) received re-irradiation however the swimmer's plot shows that 007 also was re-irradiated. Please address this discrepancy. Please also comment in the Results on the re-irradiation and if any other treatments were allowed while on-study.

Thank you for pointing this out, we have now corrected this discrepancy.

We have now added text to explain what other treatments were allowed on-trial, stating:

“After enrolling on the trial, no additional, non-protocol-directed chemotherapies, molecularly targeted therapies nor immunotherapies were allowed. Local therapies such as intratumoral cyst drainage or re-irradiation were allowed; if reirradiation was administered, subsequent infusions resumed after 4 weeks from the end of re-irradiation.”

3. The criteria for being on-study should be better described in the manuscript, not just in the clinical protocol provided in the supplement. It was unclear from Fig 1c if the blue (on-study time) represents the length of time of clinical response/improvement for each patient. Please clarify this point, and if it doesn't, it would be important to add this for all patients.

Thank you for this important comment. We have now added the clarification to Figure 1 that “Patients remained on-trial (blue) until disease progression that was unresponsive to CAR T-cell therapy.”

We include the time course of tumor volumetric changes (Figure 2, 3, Extended Data Figure 4) and the clinical improvement scores over time (Extended Data Figure 7), but given the often mixed picture of tumor volume change and clinical improvement (some patients improved clinically with no tumor size change, some patients with marked tumor reduction were transiently more symptomatic), it is difficult to represent this complex picture in the swimmer's plot.

4. Were toxicities – either CRS, ICANS or TIAN -- associated with tumor burden at time of treatment or tumor location or i.c.v. infusion dose. Please comment in the manuscript.

We have now added the comment: *“... tumors located in the lower brainstem were associated with more severe TIAN... In general, earlier infusions and larger tumors were associated with the most toxicity. The correlation of more severe TIAN with higher levels of cytokines/chemokines related to myeloid responses such as MCP-1/CCL2 raises the prospect that myeloid cells may contribute to neurological symptoms induced by CAR T-cell therapy for CNS tumors, a hypothesis that requires testing in future studies. The observed decrease in TIAN severity with repeated infusions is notable and could be due to decreased tumor burden with repeated infusions, increased immune-modulatory or immune-suppressive mechanisms in the tumor microenvironment, or both.”*

5. The volumetric change in tumor size for many of the patients, particularly the spinal DMG, are impressive. Please note what timepoint the ‘best radiographic volume change’ was achieved, possibly by indicating on the patient-specific tumor volume plots

Thank you for the helpful suggestions, we have now added this information to the graph in Figure 2b.

6. For Figure 2d (and Fig 3d and supplemental figs) it is unclear how many days after the i.c.v. infusion cytokines were measured. Is the first time point (day 43, 114, 176 etc) pre or post i.c.v. administration?

Clarification has been added to the legend: *“Timepoints expressed as days from first infusion; (first infusion always administered i.v., subsequent infusions always administered i.c.v.); infusions are separated by a vertical white spacing to indicate the beginning of a new infusion cycle. Infusion number indicated by solid bar above each heatmap.”*

7. While the authors' restraint in concluding a survival benefit is appreciated, median survival times and confidence intervals for the Kaplan Meier plots (Supp Fig 4) should be reported. It would also seem relevant to report overall survival from time of treatment.

We thank the referee for this suggestion and have now added the median survival times, 90/10 confidence intervals, and Kaplan-Meier plots illustrating the overall survival from time of treatment.

8. Are there any learnings from the cytokine changes in the plasma or CSF that can be gleaned regarding toxicity, response, or progression?

We thank the referee for this important comment. To address this important question, we examined cytokine/chemline levels that correlated with toxicity or response. Data were grouped by patient, infusion, sample type, and cytokine. We conducted statistical analyses comparing the maximum cytokine values per infusion based on toxicity (CRS Grade 0-1 vs Grade 2+; TIAN Grade 0-1 vs Grade 2+) and best response (defined based on waterfall plot, responder (tumor reduced in size) vs non-responder (tumor increased in size)) using a Mann-Whitney U Test. IV and ICV infusions were compared separately.

For cytokine release syndrome (CRS), we analyzed the data bucketing toxicity by CTCAEv5.0 Grade (Grade 0-1 vs Grade 2+). During the first infusion (IV), plasma IL-2 and MCP1 levels associated with Grade 2+ CRS (shown in Extended Data Figure 7a and below):

Extended Data Fig. a. Correlates of cytokine release syndrome (CRS). Higher plasma MCP-1 and IL2 levels correlate with patients who experiences grade 2 or higher CRS. (CRS2+, n=7 patients; CRS0-1, n= 4 patients)

For tumor inflammation-associated neurotoxicity (TIAN), we again bucketed toxicity by Grade (Grade 0-1 vs Grade 2+). All significant cytokines were associated with infusion 2+, the ICV infusions. CSF TNF α , IL10, and MCP1 were associated with Grade 2+ TIAN (Extended Data Fig. 7b and below.)

Extended Data Fig. 7b. Correlates of tumor inflammation-associated neurotoxicity (TIAN). Higher CSF levels of MCP-1, IL10 and TNFα correlate with higher grade TIAN (grade 2 and higher). (TIAN2+, n=15 patients; TIAN0-1, n= 33 patients)

For best response, responders had higher IL2 levels compared to non-responders (shown in Extended Data Fig. 7d and below.)

Extended Data Fig. 7d. Correlates of Response. Higher plasma levels of IL2 were found in patients who responded to GD2-CAR T cell therapy (decreased tumor volume at best response, n=8 patients) compared to non-responders (increased tumor volume at best response, n=3 patients).

CSF levels of immunosuppressive TGF-beta factors increased following GD2-CART administrations in some patients, and higher TGF-beta1 levels early following GD2-CART infusion trended with progression of disease (Extended Data Fig. 8a-c). Notably, TGF-beta levels in CSF were relatively low in Patient 010, who had a complete response, throughout his course (Extended Data Fig. 8a-b).

Extended Data Figure 8: CSF TGF-beta cytokine levels following GD2-CART infusions

a Heatmap of Log2FC TGF-beta levels from Day 0

b TGF-beta1 levels over time

c TGF-beta1 level with treatment or progression

Extended Data Fig. 8: CSF TGF-beta cytokine levels following GD2-CART infusions

- a. CSF TGF-beta cytokine levels after each infusion, expressed as log₂ fold change from Day 0 prior to the first infusion for the indicated patient. Timepoints expressed as days from first infusion; (first infusion always administered i.v., subsequent infusions always administered i.c.v.); infusions are separated by a vertical white spacing to indicate the beginning of a new infusion cycle. Infusion number indicated by solid bar above each heatmap. Each square represents one measurement.
- b. Absolute TGF-beta1 levels per patient over time. Dotted line represents each GD2-CART infusion for each patient. Data for Patient 012 is missing due to limited CSF sampling for that individual.
- c. TGF-beta1 levels before and after GD2-CAR T cell therapy (left, n= 10 patients) and at early timepoints following GD2-CART infusion before (“responding, n = 3 patients) and after progression (right, n = 4 patients). Statistical comparison by Mann-Whitney U Test.

Referee #2 (Remarks to the Author):

Patients with DIPG/DMG have a median overall survival of 11 months – It is a cancer of the most dismal prognosis and new treatments are urgently required. H3K27M-mutant diffuse midline gliomas (DMGs) express high levels GD2 at the cell surface, making GD2-CAR T-cells, a promising therapy. Authors have previously reported the first three patients treated at dose level #1 and this manuscript is the final clinical results.

In this second and final report from the clinical trial (NCT04196413), the authors present compelling evidence for using GD2 CAR T cell immunotherapy for DMG.

An intravenous CART cell dose was administered after lymphodepleting chemotherapy. Patients who responded positively were offered subsequent intracerebroventricular (ICV) infusions.

Thirteen patients joined the trial, and eleven received the treatment. The cells were successfully manufactured for all patients, in an average of 7 days. At the lower dose level, there were no serious side effects, but at the higher dose, three patients had a severe immune reaction called cytokine release syndrome. Nine patients also received additional doses directly into their brain fluid, which did not cause severe side effects, but 1e6/kg was established as the maximally tolerated dose IV.

All patients experienced some level of brain inflammation, but four patients had significant reductions in their tumour size, and astonishingly one patient’s tumour completely disappeared for over 30 months. Figure 2 shows patient 10’s complete response, with a disappearing tumour volume (Fig2b) and an absence of detecting H3K27M mutations in CSF via ddPCR (Fig2c). The response is quite extraordinary. For this devastating tumour type, this is an incredibly exciting result. Eight patients showed improvement in their neurological symptoms.

There was excellent supportive care, toxicity monitoring, clinical management and response assessment in place. The brain inflammation side effects were managed carefully with close monitoring and a meticulous treatment plan.

In general clinical improvement is tracked with tumour response on MRI. ICANS was rarely seen and observed only after IV administration but not with ICV infusions, which provides evidence that intracranial delivery of CAR T cells to treat brain tumours is safe. The TIAN seen was transient and manageable. Interestingly, the CAR construct contained an inducible caspase-9, but the agent available to ablate the CAR T cells via Caspase-9 induction (AP1903) was not administered to any patients treated on Arm A.

The well-written manuscript has no major flaws prohibiting its publication and it is a novel study. The statistical tests applied are sound. Overall, the treatment shows immense promise in reducing tumours and improving symptoms in these patients, a cohort who currently have no other options. Future trials will move to intracranial ICV delivery only and explore the role of lymphodepleting chemotherapy. The demonstration of such positive CART cell responses against such an insidious solid tumour in the CNS is exciting. The report is likely to be highly cited in the field.

We thank the referee for the positive and supportive comments.

Major Points:

- Given the differential responses in patients and the diversity in product delivered, it would be beneficial for authors to include, if possible, the stratification of the proportion of CD4 or CD8 cells, of the CD3+ CART+ population that persisted in the patients. Did patients #6 and #10 have a differential skewing of CD4/CD8 ratio?

We thank the referee for this interesting question. We evaluated CAR T cell product CD4/CD8 ratio in the context of patient response and identified that this was not significantly different between patients who responded (exhibited tumor shrinkage) compared to those without response (tumor growth). This analysis is now added to Extended Data Figure 7c, and is shown below.

As suggested, we also compared just patients 6 and 10 to the rest of the patients, and found no significant differences in CAR T cell product CD4/CD8 ratios.

- T cells can downregulate cell surface CAR after activation. The authors have used two methods to determine CAR T cell persistence and report PCR as a read-out for CAR T cell persistence in the main figures 2C and 3C. However, the authors measured cell surface CAR expression (as shown in Sup Fig 3e, presumably using an anti-idiotype antibody, as is described in Supp Fig 5b) – with varying results. Can the authors reconcile why the cell surface expression seems variable, and they have elected to display PCR as a readout of CAR expression? Note also the anti-CAR staining is missing from the supplementary methods.

We appreciate the referee's comment regarding understanding CAR T cell expansion in patients. We used a CAR T cell antibody staining by clone 1A7, now described in detail in the reporting summary of this paper and with reference in our methods to the detailed description in our prior publication. This staining is effective for identifying CAR T cells, but as the referee suggests, will not account for downregulation of the CAR T cell from the cell surface. In our trial, we did have many time points when CAR T cell detection was limited in flow cytometry but when we were able to detect CAR T cells by qPCR, suggesting this downregulation. To account for this and accurately represent CAR T cell expansion and persistence, we utilized qPCR, but were only able to run this assay with sufficient DNA extraction, which was only possible on peripheral blood samples.

Due to the limited CSF cell counts, we rarely had enough cells to be able to run either qPCR or flow cytometry to quantify CAR T cells over time. During timepoints when we did have enough cells, we conducted flow cytometry, as represented in the manuscript. We ran 88 CSF samples with sufficient cells for flow cytometry, of which 36 samples demonstrated presence of CAR T cells above 0.1% of CD3+ T cells in CSF. Those data are represented in Extended Data Figure 5, but are insufficient to track CAR T cell expansion and persistence in patients. A more sensitive approach to measuring and tracking CAR T cells is by qPCR, which we utilized in peripheral blood, where there was enough DNA to extract and utilize on this platform.

Minor Points:

- Supp Fig 1e. It is unclear in the figure legend what the n numbers are representing on the graph – There are 19 data points but only 11 patients were reported in this study – Where do the other 8 data points come from? Similarly, it is unclear what the data points represent in Supp Fig1D (assuming a minority of patients had CD4/CD8 ratios determined by flow at apheresis but n should be stipulated in legend for all panels. I assume Supp Fig1F is the characterisation of drug product on the day of administration but this too should be clarified in legend.

20 CAR T cell products were successfully manufactured for 13 patients. Deep characterization of 19 products are represented in the manuscript. The product for patient 11 was not characterized, as this patient developed progression and died prior to CAR T cell infusion. For the remaining 12 patients, 7 CAR T cell products were re-manufactured in order to provide repeated ICV infusions. This information has now been included in the manuscript. Thank you for pointing out the need to clarify this.

- There are some typesetting issues in the clinical trial protocol in the supplementary figures where some embedded links have been formatted incorrectly in the final PDF (Example page 41/161 and bottom of page 58) and some small typos (Page 43/161: “.... Were consistently filters when put through” Which are tiny but authors may want to correct.
- A couple of typos in the manuscript (Page 10, pseudoprogession)

Thank you – these have been fixed.

Referee #3

Monje et al. present a Phase 1 study of T cells expressing a 2nd generation GD2-specific CAR (CD8TM,41BB costim) administered first IV followed by ICV injection on two dose levels in patients with H3K27M DIPGs and sDMGs. Cells were manufactured by the Miltenyi Prodigy system in 7 days with IL7/15 and were enriched for central memory subset. The manuscript outlines safety parameters and defines the MTD. Objective responses are demonstrated by RANO2 criteria in 4/11 patients and

additional four patients had meaningful clinical benefit as measured by the team's standardized clinical improvement score. The team provides data on CAR T cell kinetics in peripheral blood and ctDNA in CSF, latter nicely demonstrating responses in addition to imaging and clinical findings. Interestingly DLTs occurred in the context of IV CAR T cell administration, but were manageable for ICV administration; thus, prompting the team to switch to ICV injections for future studies.

The manuscript is a major milestone in our understanding of CAR T cells' safety and efficacy to treat patients with DIPG/sDMG. The manuscript is overall well written, but could benefit from additional analyses as outlined below:

We thank the Referee for the positive and supportive comments.

1. The incidence and severity of CRS at 3×10^6 /kg IV dose is surprising. Table 1 summarizes these toxicities, but would benefit from more in-depth assessment and reporting.

- Can the authors provide more detail on what contributed to CRS on the IV dose? Specifically, the components of the CRS grading system would be informative to report: fever, degree of O2 requirement, pressor support. How do these IV CAR T cell mediated toxicities compare to Buffalo et al NEJM 2023 or Lin et al JCO 2024? Did the team notice peripheral nerve related toxicities after IV CAR T cell administration?

We have now added text to report that *“CRS was similar in nature to that observed in other CAR T-cell trials, with the 3 instances of grade 4 CRS characterized by hypotension requiring multiple pressors, pulmonary edema and respiratory failure requiring BiPAP/CPAP or intubation.”*

We did not observe peripheral nerve-related toxicity, aside from one patient with a painful peripheral neuropathy in a stocking-glove distribution determined to be due to B6 toxicity and that resolved with cessation of B6 supplementation. We have added this to the text, stating:

“One patient (Patient 005) experienced a Grade 2 sensory neuropathy in a stocking-glove distribution after the first infusion for which vitamin B6 toxicity was considered a contributing factor and which resolved after vitamin B6 cessation. Of note, we did not observe any cases of the transient painful peripheral neuropathy commonly associated with anti-GD2 antibody therapy³².”

- Fig 2 and 3 shows data from two selected patients for cytokines, but it would be important to gain a general view from the treated patients and compare the findings to previously reported cytokine profiles.

We have represented overall cytokine signatures of patients in Extended Data Fig. 6, and we have now added that these signatures are similar to the DL1 patient cytokine signatures previously reported in Majzner et al. 2022. We have also added more cytokine analyses of correlations to toxicity and response to Extended Data Figure 7.

- Supplementary table 2 outlines TIAN mgmt but is missing information on CRS mgmt; the inducible caspase 9 was fortunately not needed. What mgmt strategies were employed for CRS (some may overlap with TIAN/ICANs)?

We have now added text to describe CRS management: *“CRS was similar in nature to that observed in other CAR T-cell trials¹⁴. The three instances of Grade 4 CRS were characterized by hypotension requiring multiple pressors (one patient), pulmonary edema and respiratory failure requiring BiPAP/CPAP (one patient) or intubation (one patient). CRS was managed according to established guidelines using tocilizumab, anakinra, corticosteroids (dexamethasone and methylprednisolone), fluid resuscitation, and supportive care.”*

- Are there differences in specific CSF cytokine/chemokine levels in responders vs non-responders?

We thank the referee for this important comment. To address this important question, we examined cytokine/chemokine levels that correlated with toxicity or response. Data were grouped by patient, infusion, sample type, and cytokine. We conducted statistical analyses comparing the maximum cytokine values per infusion based on toxicity (CRS Grade 0-1 vs Grade 2+; TIAN Grade 0-1 vs Grade 2+) and best response (defined based on waterfall plot, responder (tumor reduced in size) vs non-responder (tumor increased in size)) using a Mann-Whitney U Test. IV and ICV infusions were compared separately. For cytokine release syndrome (CRS), we analyzed the data bucketing toxicity by CTCAEv5.0 Grade (Grade 0-1 vs Grade 2+). During the first infusion (IV), plasma IL-2 and MCP1 levels associated with Grade 2+ CRS (shown in Extended Data Figure 7a and below.)

Extended Data Fig. a. Correlates of cytokine release syndrome (CRS). Higher plasma MCP-1 and IL2 levels correlate with patients who experiences grade 2 or higher CRS. (CRS2+, n=7 patients; CRS0-1, n= 4 patients)

For tumor inflammation-associated neurotoxicity (TIAN), we again bucketed toxicity by Grade (Grade 0-1 vs Grade 2+). All significant cytokines were associated with infusion 2+, the ICV infusions. CSF TNF α , IL10, and MCP1 were associated with Grade 2+ TIAN (Extended Data Fig. 7b and below.)

Extended Data Fig. 7b. Correlates of tumor inflammation-associated neurotoxicity (TIAN). Higher CSF levels of MCP-1, IL10 and TNF α correlate with higher grade TIAN (grade 2 and higher). (TIAN2+, n=15 patients; TIAN0-1, n= 33 patients)

For best response, responders had higher IL2 levels compared to non-responders (shown in Extended Data Fig. 7d and below.)

Extended Data Fig. 7d. Correlates of Response. Higher plasma levels of IL2 were found in patients who responded to GD2-CAR T cell therapy (decreased tumor volume at best response, n=8 patients) compared to non-responders (increased tumor volume at best response, n=3 patients).

To understand patient progression, we compared the first ICV infusion to the first progression infusion in each patient for whom we had data. Due to the small sample size, the number of patients within this analysis was limited, and maximum cytokine level was compared between cohorts. Peak levels of IP10, a chemokine secreted in response to interferon-gamma signaling, were decreased at the time of progression compared to timepoints during response in the same patients; dotted line indicates paired values for individual patients at time of response (first ICV infusion) and time of progression. (Shown in **Extended Data Fig. 7e** and below:)

Extended Data Figure 7e. Peak levels of IP10, a chemokine secreted in response to interferon-gamma signaling, were decreased at the time of progression compared to timepoints during response in the same patients; dotted line indicates paired values for individual patients (n=3 patients with paired samples) at time of response (first ICV infusion) and time of progression.

CSF levels of immunosuppressive TGF-beta factors increased following GD2-CART administrations in some patients, and higher TGF-beta1 levels early following GD2-CART infusion trended with progression of disease (Extended Data Fig. 8a-c). Notably, TGF-beta levels in CSF were relatively low in Patient 010, who had a complete response, throughout his course (Extended Data Fig. 8a-b).

Extended Data Figure 8: CSF TGF-beta cytokine levels following GD2-CART infusions

a Heatmap of Log2FC TGF-beta levels from Day 0

b TGF-beta1 levels over time

c TGF-beta1 level with treatment or progression

Extended Data Fig. 8: CSF TGF-beta cytokine levels following GD2-CART infusions

a. CSF TGF-beta cytokine levels after each infusion, expressed as log2 fold change from Day 0 prior to the first infusion for the indicated patient. Timepoints expressed as days from first infusion; (first infusion always administered i.v., subsequent infusions always administered i.c.v.); infusions are separated by a vertical white spacing to indicate the beginning of a new infusion cycle. Infusion number indicated by solid bar above each heatmap. Each square represents one measurement.

b. Absolute TGF-beta1 levels per patient over time. Dotted line represents each GD2-CART infusion for each patient. Data for Patient 012 is missing due to limited CSF sampling for that individual.

c. TGF-beta1 levels before and after GD2-CAR T cell therapy (left, n= 10 patients) and at early timepoints following GD2-CART infusion before (“responding, n = 3 patients) and after progression (right, n = 4 patients). Statistical comparison by Mann-Whitney U Test.

2. On the consort diagram, 83 patients were evaluated, 34 were excluded due to not meeting eligibility criteria and 13 were enrolled. Please, clarify what happened to patients meeting eligibility criteria, but not enrolled.

The remaining 36 patients chose other trials, chose not to enroll on a clinical trial, or progressed to the point of ineligibility while on the waitlist. This clarification has been added to the consort diagram legend.

3. CAR T cell expansion and persistence related assessments:

- Some of the peripheral blood kinetics are shown in Fig 2 & 3 (and in suppl figs), but these are two selected patients and it is important to provide an overview of this critical parameter for all patients as a main figure. A similar graph with ctDNA would be highly informative especially if expansion/persistence can be correlated with drop in ctDNA.

We agree with the reviewer that this is critical to understand the trend across all patients. Complete ctDNA and qPCR data is represented in Extended Data Figure 5.

In order to assess for a correlation of peripheral blood GD2 CAR T cells by qPCR and ctDNA tumor in CSF, we conducted a linear regression analysis for the two patients for whom we had adequate paired timepoints and ctDNA increase represented. Linear regression analysis identified a trending negative association of these two parameters in Patient 006 (slope = -0.2975) and Patient 009 (slope = -0.0004527). As we and other trials treat patients, this will be an important marker to further explore and understand. We include this in referee responses for the sake of discussion and look forward to evaluating this as more patients and data are accumulated in the future.

- What is the expansion and persistence of CAR T cells in CSF? This parameter seems to be missing from the manuscript.

We appreciate the referee’s comment regarding the importance of understanding CAR T cell expansion in CSF. Unfortunately, due to the limited CSF cell counts, we rarely had enough cells to be able to run either qPCR or flow cytometry to quantify CAR T cells over time. During timepoints when we did have enough

cells, we prioritized flow cytometry, as represented in the manuscript. We ran 88 CSF samples with enough cells available for flow cytometry, of which 36 samples demonstrated presence of CAR T cells above 0.1% of CD3+ T cells in CSF. Those data are represented in Extended Data Figure 5, but are insufficient to track CAR T cell expansion and persistence in patients.

4. One of the key findings from Majzner et al was the demonstration of antitumor activity of ICV dosing. With the number of patients treated now, can the authors summarize and comment on benefit of repeat infusions after IV dosing (beyond increased safety).

We have designed ICV-only arms of the trial to test if ICV-only therapy is equally effective as IV followed by repeated ICV, so it is too early to make a definitive assessment. That said, we have certainly seen efficacy with repeated ICV infusions in this cohort of patients and we have made a stronger statement about this in the discussion.

Minor:

Could you clarify the number of patients with clinical and imaging responses. The abstract lists four + four, but the text states “nine experienced clinical or imaging benefit”.

Thank you for pointing this out. 9 patients experienced clinical benefit and of these, 4 had major tumor reductions; we have now corrected the discrepancy.

The product is enriched for CD4 cells although the cytokines used should promote CD8s. Is CD4 promoting effect related to dasatinib?

This is a great question. We do not attribute this to dasatinib as we see this consistently using CD3/CD28-based T cell activation and manufacturing in the Miltenyi Prodigy as previously reported in Spiegel et al (Nature Med, PMID: 34312556) and in Frank et al (Lancet, PMID: 38996463) where dasatinib was not used.

Suppl. Fig 1e and 3: there are more measurements on the graphs than the number of patients. Are some of these duplicates? Please, clarify.

20 CAR T cell products were successfully manufactured for 13 patients. Deep characterization of 19 products are represented in the manuscript. The product for patient 11 was not characterized, as this patient developed progression and died prior to CAR T cell infusion. For the remaining 12 patients, 7 CAR T cell products were re-manufactured in order to provide repeated ICV infusions. This information has now been included in the manuscript for clarity.

The manuscript text states max infusion of 16, but the figure 1 seems to show 17. Please, clarify.

Thank you for pointing out this discrepancy; 17 is correct and this has now been fixed.

What do the asterisks represent in Fig 1c?

Asterisks indicates time of trial enrollment, with first treatment indicated by the y-axis at time 0.

Reviewer Reports on the First Revision:

Referees' comments:

Referee #1 (Remarks to the Author):

The authors have sufficiently addressed my comments, and I look forward to seeing this important clinical study published.

Referee #2 (Remarks to the Author):

I have now reviewed the resubmission. The authors have sufficiently addressed all of the points raised comprehensively and thoughtfully. I have no further comments on the manuscript.

Referee #3 (Remarks to the Author):

The authors addressed the comments and prepared an even more informative manuscript. The work is suitable for publication.